# Slicing the Gaussian Mixture Wasserstein Distance

**Moritz Piening**                                              *piening@math.tu-berlin.de*
*Institut für Mathematik*
*Technische Universität Berlin*

**Robert Beinert**                                              *beinert@math.tu-berlin.de*
*Institut für Mathematik*
*Technische Universität Berlin*

**Reviewed on OpenReview:** *https://openreview.net/forum?id=yPBtJ4JPwi*

## Abstract

Gaussian mixture models (GMMs) are widely used in machine learning for tasks such as clustering, classification, image reconstruction, and generative modelling. A key challenge in working with GMMs is defining a computationally efficient and geometrically meaningful metric. The mixture Wasserstein (MW) distance adapts the Wasserstein metric to GMMs and has been applied in various domains, including domain adaptation, dataset comparison, and reinforcement learning. However, its high computational cost—arising from repeated Wasserstein distance computations involving matrix square root estimations and an expensive linear program—limits its scalability to high-dimensional and large-scale problems. To address this, we propose multiple novel slicing-based approximations to the MW distance that significantly reduce computational complexity while preserving key optimal transport properties. From a theoretical viewpoint, we establish several weak and strong equivalences between the introduced metrics, and show the relations to the original MW distance and the well-established sliced Wasserstein distance. Furthermore, we validate the effectiveness of our approach through numerical experiments, demonstrating computational efficiency and applications in clustering, perceptual image comparison, and GMM minimization.

## 1 Introduction

Gaussian mixture models (GMMs) are a fundamental tool in machine learning and statistics; widely used in applications such as classification (Wan et al., 2019), clustering (Zhang et al., 2021; Adipoetra & Martin, 2025), image reconstruction (Zoran & Weiss, 2011; Nguyen et al., 2023; Piening et al., 2024), 3d surface representation (Zou & Sester, 2024), and generative modeling (Hagemann & Neumayer, 2021; Alberti et al., 2024). That is why defining a computationally efficient and geometrically meaningful metric on the space of GMMs is valuable for many data-driven applications. The most prominent candidate for such tasks is the mixture Wasserstein (MW) distance (Chen et al., 2016; 2018; Delon & Desolneux, 2020), which can be interpreted as restriction of the classical Wasserstein distance to GMM transport plans. The main limitation of the MW distance is its high computational cost. It consists of an 'inner' and an 'outer' optimal transport problem. The inner problem requires the computation of all pairwise Wasserstein distances between the components of the given GMMs. Despite having a closed-form solution, this step involves matrix square root calculations, which becomes highly costly for high-dimensional data. The outer problem consists in solving an optimal transport problem, whose cost matrix is determined by the inner problem. If both GMMs have $K$ Gaussian components, the minimization of the outer problem has a complexity of $\mathcal{O}(K^3)$ for exact solutions and $\mathcal{O}(K^2 \log(K))$ for approximate solutions (Peyré & Cuturi, 2019). For this reason, the MW distance becomes prohibitive for large-scale applications with high component numbers. To address these challenges, we propose an acceleration approach leveraging projection techniques. Slicing the inner and outer optimal transport problems, i.e., applying a double slicing, we significantly reduce the computational burden while preserving the geometric properties of the MW distance. Additionally, our novel sliced MW

approach facilitates fast gradient-based optimization in the space of GMMs; making the resulting metric particularly useful for machine learning applications.

## 1.1 Related work

**Sliced probability metrics**   A fundamental task in machine learning and computer vision is the meaningful comparison of high-dimensional empirical data distributions. Geometrically meaningful metrics such as the Wasserstein distance (Santambrogio, 2015; Peyré & Cuturi, 2019) are often computationally costly; therefore, sliced probability divergences are increasingly popular. In case of the classical Wasserstein distance, the underlying high-dimensional transport problem is replaced by multiple 1d transport problems. At the core, the 1d problems reduce to efficiently solvable assignment formulations (Bonneel et al., 2015; Nadjahi et al., 2020), which surmount the original, expensive linear program. The reduced computational cost makes the obtained sliced Wasserstein (SW) distance a valuable tool for image restoration (Tartavel et al., 2016), generative modeling (Deshpande et al., 2018; Liutkus et al., 2019), GMM estimation (Kolouri et al., 2018), and small-data classification tasks (Aldroubi et al., 2021; Moosmüller & Cloninger, 2023; Beckmann et al., 2025). Beyond the classical Wasserstein distances on Euclidean spaces, the slicing idea can be applied to more general divergences (Kolouri et al., 2022), the Stein discrepancy (Gong et al., 2021), the maximum mean discrepancy (Hertrich et al., 2024; Hagemann et al., 2024), and the Wasserstein distance on the sphere (Quellmalz et al., 2023; 2024).

**Metrics between probability mixtures**   The most prominent mixtures of parametrized probability measures are GMMs. Since the Wasserstein distance between GMMs with more than one component cannot be computed in closed form, an alternative metric—the MW distance—has been introduced (Delon & Desolneux, 2020; Chen et al., 2018; 2016). The MW distance has been successfully applied in various domains, including the quality assessment of GMMs (Farnia et al., 2023), domain adaptation (Montesuma et al., 2024), perceptual image evaluation (Luzi et al., 2023), single-cell data analysis (Lin et al., 2023), and reinforcement learning (Ziesche & Rozo, 2023). Beyond GMMs, the MW distance has been extended to mixtures of non-Gaussian probability measures (Alvarez-Melis & Fusi, 2020; Bing et al., 2022; Dusson et al., 2023; Wilson et al., 2024). The extended versions have, in particular, applications in the comparison of labeled datasets (Alvarez-Melis & Fusi, 2020). Upon completing our work, we became aware of two closely related preprints on sliced optimal transport distances between mixtures of probability measures (Nguyen & Mueller, 2025; Nguyen et al., 2025), which were developed independently and in parallel to our study. Notably, the proposed metrics for GMMs differ from our sliced MW variants. Moreover, we provide the first theoretical comparison of sliced MW versions with the original MW and SW distance. In particular, we establish forms of metric equivalence.

## 1.2 Contribution

The main contributions of this paper, which are also schematically visualized in Figure 1, are as follows:

- Starting from the original MW distance, we propose (i) to use the SW distance for the inner transport problem leading to the novel mixture sliced Wasserstein (MSW) distance and (ii) to slice the entire MW distance using 1d projections, which results in a sliced mixture Wasserstein (SMW) distance. For the efficient computation of the latter, we propose a further slicing based on the identification of 1d GMMs with 2d empirical measures. This procedure yields the new double-sliced mixture Wasserstein (DSMW) distance. The new distances are closely related and significantly reduce the computational complexity, while maintaining a background in optimal transport. Moreover, they are well-defined metrics on the space of all GMMs; see Theorem 3.1.

- Providing a theoretical analysis, we show that, under mild conditions, the MW, MSW, SMW, and DSMW distances are weakly equivalent, i.e., convergence of a sequence in one distance implies the convergence in all others; see Theorem 3.4. Under the same mild conditions, we show that the SMW and DSMW distances are actually strongly equivalent, i.e., the distances can be estimated by each other; see Theorem 3.2. Unfortunately, this equivalence cannot be extended to the MW and MSW distance; see Theorem 3.3.

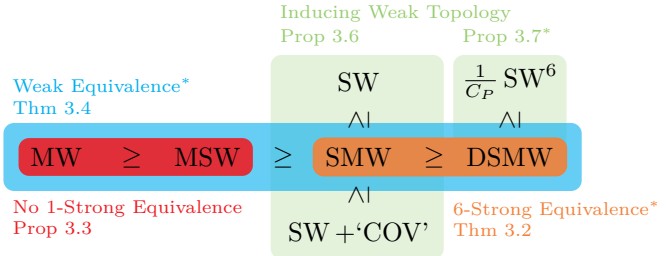

Figure 1: Relations between the considered GMM metrics; more precisely, between the proposed mixture sliced Wasserstein (MSW), sliced mixture Wasserstein (SMW), double sliced mixture Wasserstein (DSMW) distance and the original mixture Wasserstein (MW) and sliced Wasserstein (SW) distance. The SMW distance can be bounded from above by the SW distance and a non-vanishing additive term 'COV' depending on the involved covariances. Under mild conditions, the DSMW distance also induces the SW distance with bounding constant $C_P$ depending on the considered subset of GMMs.

- We compare the introduced metrics with the original SW distance. More precisely, we show that our MSW and SMW distances are upper bounds for the SW distance. Consequently, the MSW and SMW distance introduce the usual weak convergence of measures (restricted to the space of GMMs); see Proposition 3.6. For suitable subsets of GMMs, this property can also be established for the DSMW distance; see Proposition 3.6.

- The efficiency and scalability of our slicing approaches are demonstrated through comprehensive experiments that focus on a runtime comparison especially for high-dimensional GMMs with large numbers of components and possible practical applications of the new distances.

### 1.3 Outline

The remainder of this paper is structured as follows: In Section 2, we review optimal transport distances, especially, the original Wasserstein distance and its acceleration through the sliced Wasserstein distance. Furthermore, we review the original MW distance for GMMs. In Section 3, we introduce our novel slicing-based GMM metrics and provide theoretical results about the weak and strong equivalences. Moreover, we study the relations of our distances to the MW and SW distance. In Section 4, we discuss the numerical implementation and computational performance of our slicing procedures. In Section 5, the practical potential for clustering, perceptual evaluation, and gradient-based optimization is evaluated. Finally, Section 6 summarizes our findings and discusses future directions.

## 2 Preliminaries

### 2.1 Wasserstein and sliced Wasserstein distance

Optimal transport-based metrics like the Wasserstein distance and its variants gauge the similarity between different probability measures living on a common measurable space. In the following, we restrict ourselves to measures on Euclidean spaces. More precisely, for any $X \subset \mathbb{R}^d$, the space of all Borel probability measures on $X$ with respect to the Euclidean metric is denoted by $\mathcal{P}(X)$. The subset of *probability measures with finite pth moment* is defined by

$$\mathcal{P}_p(X) := \left\{ \mu \in \mathcal{P}(X) \,\Big|\, \int_X |x|^p \, \mathrm{d}\mu(x) < \infty \right\}, \quad 1 \leq p < \infty.$$

A measure $\mu \in \mathcal{P}(X)$ is transferred to another domain $X' \subset \mathbb{R}^{d'}$ via a mapping $T \colon X \to X'$ by it's *push-forward*: $T_\sharp \mu := \mu \circ T^{-1}$.

A transport plan between $\mu_0, \mu_1 \in \mathcal{P}_p(X)$ is a probability measure $\gamma \in \mathcal{P}(X \times X)$ whose marginals coincide with the given probabilities. Mathematically, based on the canonical projections onto the $i$th component given by $\pi_i(x_0, x_1) \coloneqq x_i$, the set of all *transport plans* between $\mu_1$ and $\mu_2$ reads as

$$\Gamma(\mu_0, \mu_1) \coloneqq \big\{\gamma \in \mathcal{P}(X \times X) \mid \pi_{0,\sharp}\gamma = \mu_0, \pi_{1,\sharp}\gamma = \mu_1\big\}.$$

Based on these plans, the *p-Wasserstein distance*—also known as *p-Kantorovich–Rubinstein metric*—between $\mu_0, \mu_1 \in \mathcal{P}_p(\mathbb{R}^d)$ is given by

$$\mathrm{W}_p(\mu_0, \mu_1) \coloneqq \inf_{\gamma \in \Gamma(\mu_0, \mu_1)} \Big(\int_{\mathbb{R}^d \times \mathbb{R}^d} \|x_0 - x_1\|^p \, \mathrm{d}\gamma(x_0, x_1)\Big)^{\frac{1}{p}}. \tag{1}$$

The Wasserstein distance defines a metric; so $(\mathcal{P}_p(\mathbb{R}^d), \mathrm{W}_p)$ becomes a metric space (Villani, 2003; Santambrogio, 2015).

For dimension $d > 1$, the computation of the Wasserstein distance is, in general, numerically challenging. In the special case $d = 1$, the Wasserstein distance can, however, be calculated analytically (Villani, 2003). On the basis of the *cumulative distribution function*:

$$F_\mu(x) \coloneqq \mu\big((-\infty, x]\big), \quad x \in \mathbb{R},$$

and its *generalized inverse*:

$$F_\mu^{-1}(t) \coloneqq \inf\big\{x \in \mathbb{R} \mid F_\mu(x) > t\big\}, \quad t \in (0, 1),$$

the Wasserstein distance has the closed-form solution:

$$\mathrm{W}_p(\mu_0, \mu_1) = \Big(\int_0^1 |F_{\mu_0}^{-1}(t) - F_{\mu_1}^{-1}(t)|^p \, \mathrm{d}t\Big)^{\frac{1}{p}}. \tag{2}$$

For empirical measure, the generalized inverse may be calculated using efficient sorting algorithms (Bonneel et al., 2015; Nadjahi et al., 2020).

In order to exploit the computational benefits of the one-dimensional Wasserstein distance, the so-called sliced Wasserstein distance is introduced in (Kolouri et al., 2019; Bonneel et al., 2015; Kolouri et al., 2016), which is closely related to the Radon transform. Exploiting the *slicing operator*:

$$\pi_\theta \colon \mathbb{R}^d \to \mathbb{R} : x \mapsto \theta \cdot x, \quad \theta \in \mathbb{S}^{d-1},$$

where $\mathbb{S}^{d-1} \coloneqq \{x \in \mathbb{R}^d \mid \|x\| = 1\}$ denotes the sphere, and $\bullet \cdot \bullet$ the Euclidean inner product, the *sliced p-Wasserstein distance* between $\mu_0, \mu_1 \in \mathcal{P}_p(\mathbb{R}^d)$ is defined as

$$\mathrm{SW}_p(\mu_0, \mu_1) \coloneqq \Big(\int_{\mathbb{S}^{d-1}} \mathrm{W}_p^p(\pi_{\theta,\sharp}\,\mu_0, \pi_{\theta,\sharp}\,\mu_1) \, \mathrm{d}\theta\Big)^{\frac{1}{p}}.$$

The spherical integral can be approximated using Monte Carlo methods (Bonneel et al., 2015; Nadjahi et al., 2020) or Quasi-Monte Carlo methods (Nguyen et al., 2024; Hertrich et al., 2025).

From a topological point of view, the sliced Wasserstein and Wasserstein distance are closely related. On the one side, $\mathrm{SW}_p(\mu_0, \mu_1) \leq \mathrm{W}_p(\mu_0, \mu_1)$ for any $\mu_0, \mu_1 \in \mathcal{P}_p(\mathbb{R}^d)$. On the other side, for compact subsets $X \subset \mathbb{R}^d$, both distances are $p(d+1)$-*strongly equivalent* (Bonnotte, 2013), i.e., there exists a constant $C_X > 0$ such that

$$\mathrm{SW}_p(\mu_0, \mu_1) \leq \mathrm{W}_p(\mu_0, \mu_1) \leq C_X \, \mathrm{SW}_p^{\frac{1}{p(d+1)}}(\mu_0, \mu_1) \quad \forall \mu_0, \mu_1 \in \mathcal{P}(X). \tag{3}$$

Beyond compact sets, the sliced Wasserstein and Wasserstein distance are *weakly equivalent* (Nadjahi et al., 2019; 2020), i.e., $\mathrm{SW}_p(\mu_n, \mu) \to 0$ is equivalent to $\mathrm{W}_p(\mu_n, \mu) \to 0$ as $n \to \infty$ for any sequence $(\mu_n)_{n \in \mathbb{N}} \subset \mathcal{P}_p(\mathbb{R}^d)$ and any measure $\mu \in \mathcal{P}_p(\mathbb{R}^d)$. Moreover, the convergence under W and SW imply *weak convergence* $\mu_n \rightharpoonup \mu$ in $\mathcal{P}(\mathbb{R}^d)$ (Nadjahi et al., 2019; 2020), i.e., for every continuous function $\phi \colon \mathbb{R}^d \to \mathbb{R}$ vanishing at infinity, it holds

$$\lim_{n \to \infty} \int_{\mathbb{R}^d} \phi(x) \, \mathrm{d}\mu_n(x) = \int_{\mathbb{R}^d} \phi(x) \, \mathrm{d}\mu(x).$$

## 2.2 Wasserstein distance between Gaussian mixtures

Computing the Wasserstein distance between arbitrary probability distributions is numerically challenging, but, in some specific cases, closed-form solutions are available (Santambrogio, 2015). One such instance is the 2-Wasserstein distance between two Gaussian distributions $\mu_0 \sim \mathcal{N}(m_0, \Sigma_0)$ and $\mu_1 \sim \mathcal{N}(m_1, \Sigma_1)$ on $\mathbb{R}^d$ with means $m_i \in \mathbb{R}^d$ and covariance matrices $\Sigma_i \in S_d^+$, which are symmetric and positive semi-definite in $\mathbb{R}^{d \times d}$. For these measures, we have the explicit formula (Delon & Desolneux, 2020):

$$\mathrm{W}_2^2(\mu_0, \mu_1) = \|m_0 - m_1\|^2 + \mathrm{tr}\big(\Sigma_0 + \Sigma_1 - 2\big(\Sigma_0^{\frac{1}{2}} \Sigma_1 \Sigma_0^{\frac{1}{2}}\big)^{\frac{1}{2}}\big). \tag{4}$$

On the real line with $\mu_i \sim \mathcal{N}(m_i, \sigma_i^2)$, this simplifies to

$$\mathrm{W}_2^2(\mu_0, \mu_1) = (m_0 - m_1)^2 + (\sigma_0 - \sigma_1)^2 = \|(m_0, \sigma_0) - (m_1, \sigma_1)\|_2^2. \tag{5}$$

Unfortunately, there exists no explicit formula for Gaussian mixtures. Restricting the transport plans $\gamma$ in (1) to Gaussian mixtures as well, Delon and Desolneux (Delon & Desolneux, 2020) propose an adapted Wasserstein metric that explicitly exploit the closed forms (4) and (5). More precisely, a *finite Gaussian mixture model* (GMM) has the form

$$\mu := \sum_{k=1}^{K} \omega^k \mu^k, \quad \mu^k \sim \mathcal{N}(m^k, \Sigma^k), \tag{6}$$

with weights $\omega := (\omega^k)_{k=1}^K$ in the *probability simplex* $\Delta_K := \{\omega \in \mathbb{R}_{\geq 0}^K \mid \omega \cdot \mathbf{1}_K = 1\}$, where $\mathbf{1}_K \in \mathbb{R}^K$ denotes the all-ones vector. The set of Gaussian mixtures on $\mathbb{R}^d$ with at most $K$ components is denoted by $\mathrm{GMM}_d(K)$. Obviously, $\mathrm{GMM}_d(K) \subset \mathrm{GMM}_d(K')$ for $K < K'$. The collection of all finite Gaussian mixtures is defined by

$$\mathrm{GMM}_d(\infty) = \bigcup_{K>0} \mathrm{GMM}_d(K).$$

Using the discrete transport plans between $\omega_0 \in \Delta_{K_0}$ and $\omega_1 \in \Delta_{K_1}$ that are defined by

$$\Gamma(\omega_0, \omega_1) := \big\{\gamma \in \mathbb{R}_{\geq 0}^{K_0 \times K_1} \mid \gamma \, \mathbf{1}_{K_1} = \omega_0, \gamma^\top \mathbf{1}_{K_0} = \omega_1\big\},$$

the *mixture (2-)Wasserstein* (MW) *distance* (Delon & Desolneux, 2020; Chen et al., 2016; 2018) between $\mu_0 \in \mathrm{GMM}_d(K_0)$ and $\mu_1 \in \mathrm{GMM}_d(K_1)$ with components as in (6) reads as

$$\mathrm{MW}(\mu_0, \mu_1) := \min_{\gamma \in \Gamma(\omega_0, \omega_1)} \Big(\sum_{k_0=1}^{K_0} \sum_{k_1=1}^{K_1} \gamma_{k_0, k_1} \, \mathrm{W}_2^2(\mu_0^{k_0}, \mu_1^{k_1})\Big)^{\frac{1}{2}}. \tag{7}$$

The MW distance is a geodesic metric on $\mathrm{GMM}_d(\infty)$ that is neither strongly nor weakly equivalent to the 2-Wasserstein distance (Delon & Desolneux, 2020). However, (Delon & Desolneux, 2020, Prop. 6) establishes the lower and upper bounds

$$\mathrm{W}_2(\mu_0, \mu_1) \leq \mathrm{MW}(\mu_0, \mu_1)$$
$$\leq \mathrm{W}_2(\mu_0, \mu_1) + \sqrt{2 \sum_{k_0=1}^{K_0} \omega_0^{k_0} \, \mathrm{tr}(\Sigma_0^{k_0})} + \sqrt{2 \sum_{k_1=1}^{K_1} \omega_1^{k_1} \, \mathrm{tr}(\Sigma_1^{k_1})}. \tag{8}$$

On the real line, the MW distance may be interpreted as (classical) Wasserstein distance on the parameter space $\mathbb{R} \times \mathbb{R}_{\geq 0}$ of the Gaussian distributions. Mapping the parameters of $\mu \in \mathrm{GMM}(K)$ to the point measure

$$\nu(\mu) = \sum_{k=1}^{K} \omega^k \, \delta_{(m^k, \sigma^k)},$$

where $\delta_\bullet$ denotes the Dirac measure, and using (5), we obtain

$$
\begin{aligned}
\mathrm{MW}(\mu_0, \mu_1) &= \min_{\gamma \in \Gamma(\omega_0, \omega_1)} \Big( \sum_{k_0=1}^{K_0} \sum_{k_1=1}^{K_1} \gamma_{k_0, k_1} \left\| (m_0^{k_0}, \sigma_0^{k_0}) - (m_1^{k_1}, \sigma_1^{k_1}) \right\|^2 \Big)^{\frac{1}{2}} \\
&= \mathrm{W}_2(\nu(\mu_0), \nu(\mu_1))
\end{aligned}
$$

for all $\mu_0 \in \mathrm{GMM}_1(K_0)$ and $\mu_1 \in \mathrm{GMM}_1(K_1)$.

## 3 Sliced distances for Gaussian mixtures

From a computational point of view, there are two main disadvantages of the MW distance: 1. The computation of the Wasserstein distance between two Gaussians for $d > 1$ requires the calculation of matrix square roots and, therefore, of costly eigenvalue decompositions. 2. The outer optimal transport between the Gaussian components becomes expensive for large numbers of components, even when using entropic regularization (Cuturi, 2013). To overcome both issues, we propose different slicings of the MW distance.

### 3.1 Mixture sliced and sliced mixture distances

As a remedy to the first disadvantage—the computation of Wasserstein distances between Gaussians—, we define the *mixture sliced Wasserstein* (MSW) *distance*:

$$
\mathrm{MSW}^2(\mu_0, \mu_1) := \min_{\gamma \in \Gamma(\omega_0, \omega_1)} \sum_{k_0=1}^{K_0} \sum_{k_1=1}^{K_1} \gamma_{k_0, k_1} \, \mathrm{SW}_2^2(\mu_0^{k_0}, \mu_1^{k_1}),
$$

where $\mu_0 \in \mathrm{GMM}(K_0)$ and $\mu_1 \in \mathrm{GMM}(K_1)$ have form (6). The MSW distance is especially useful for GMMs with few high-dimensional components, where the slicing yields a significant speed-up, and where we are interested in recovering a transport plan between Gaussian components. However, it still requires solving an optimal transport problem, which becomes problematic for high numbers of components, and we therefore aim for a 'fully sliced' distance.

As the first step in this direction, we instead consider the *sliced mixture Wasserstein* (SMW) *distance*:

$$
\mathrm{SMW}^2(\mu_0, \mu_1) := \int_{\mathbb{S}^{d-1}} \mathrm{MW}^2(\pi_{\theta, \sharp} \mu_0, \pi_{\theta, \sharp} \mu_1) \, \mathrm{d}\theta = \int_{\mathbb{S}^{d-1}} \mathrm{W}_2^2(\nu_\theta(\mu_0), \nu_\theta(\mu_1)) \, \mathrm{d}\theta,
$$

where $\nu_\theta(\mu_i) := \nu(\pi_{\theta, \sharp} \mu_i)$. Since $\pi_{\theta, \sharp} \mu_i \in \mathrm{GMM}_1(K_i)$, the SMW distance is well-defined. However, this does not offer computational advantages as it requires solving a non-trivial 2d optimal transport problem for each projection. To speed up the computation, we switch to the sliced Wasserstein distance yielding the *double-sliced mixture Wasserstein* (DSMW) *distance*:

$$
\mathrm{DSMW}^2(\mu_0, \mu_1) := \int_{\mathbb{S}^{d-1}} \mathrm{SW}_2^2(\nu_\theta(\mu_0), \nu_\theta(\mu_1)) \, \mathrm{d}\theta.
$$

This distance solves both computational issues since we have neither to deal with Wasserstein distance between Gaussians nor to minimize an optimal transport problem. Instead, we can exploit the closed-form solution (2) to compute the required Wasserstein distances between point measures on the line.

**Theorem 3.1.** *MSW, SMW, and DSMW are metrics on* $\mathrm{GMM}_d(\infty)$ *satisfying*

$$
\mathrm{DSMW}(\mu_0, \mu_1) \leq \mathrm{SMW}(\mu_0, \mu_1) \leq \mathrm{MSW}(\mu_0, \mu_1) \leq \mathrm{MW}(\mu_0, \mu_1), \qquad \mu_0, \mu_1 \in \mathrm{GMM}(\infty). \tag{9}
$$

*Proof.* For any $\mu_i \in \mathrm{GMM}(\infty)$, $i = 0, 1$, there exists $K_i \in \mathbb{N}$ such that $\mu_i \in \mathrm{GMM}(K_i)$. The first and last inequalities directly follow from $\mathrm{SW}_2(\mu_0, \mu_1) \leq \mathrm{W}_2(\mu_0, \mu_1)$. The remaining inequality can be established by

$$\mathrm{SMW}^2(\mu_0, \mu_1) = \int_{\mathbb{S}^{d-1}} \min_{\gamma \in \Gamma(\omega_0, \omega_1)} \sum_{k_0=1}^{K_0} \sum_{k_1=1}^{K_1} \gamma_{k_0, k_1} \mathrm{W}_2^2(\pi_{\theta, \sharp} \mu_0^{k_0}, \pi_{\theta, \sharp} \mu_1^{k_1}) \, \mathrm{d}\theta$$

$$\leq \min_{\gamma \in \Gamma(\omega_0, \omega_1)} \sum_{k_0=1}^{K_0} \sum_{k_1=1}^{K_1} \gamma_{k_0, k_1} \int_{\mathbb{S}^{d-1}} \mathrm{W}_2^2(\pi_{\theta, \sharp} \mu_0^{k_0}, \pi_{\theta, \sharp} \mu_1^{k_1}) \, \mathrm{d}\theta = \mathrm{MSW}^2(\mu_0, \mu_1).$$

The positivity and symmetry of all distances are clear. The triangle inequalities for SMW and DSMW directly follow from the triangle inequalities for $\mathrm{W}_2$ and $\mathrm{SW}_2$ together with the triangle inequality for the 2-norm on $\mathbb{S}^{d-1}$. To show the triangle inequality for MSW, we consider $\mu_i \in \mathrm{GMM}_d(K_i)$, $i = 0, 1, 2$. Let $\gamma^{0,1,*}$ and $\gamma^{1,2,*}$ realize $\mathrm{MSW}(\mu_0, \mu_1)$ and $\mathrm{MSW}(\mu_1, \mu_2)$ respectively, and choose $\eta \in \mathbb{R}_{\geq 0}^{K_0 \times K_1 \times K_2}$ such that

$$\sum_{k_2=1}^{K_2} \eta_{k_0, k_1, k_2} = \gamma_{k_0, k_1}^{0,1,*} \quad \text{and} \quad \sum_{k_0=1}^{K_0} \eta_{k_0, k_1, k_2} = \gamma_{k_1, k_2}^{1,2,*}.$$

The existence of such $\eta$ is guaranteed by the so-called gluing lemma (Villani, 2003, Lem 7.6). Since $\gamma^{0,2}$ given by $\gamma_{k_0, k_2}^{0,2} := \sum_{k_1=1}^{K_1} \eta_{k_0, k_1, k_2}$ satisfies $\gamma^{0,2} \in \Gamma(\omega_0, \omega_2)$, we obtain

$$\mathrm{MSW}(\mu_0, \mu_2) \leq \sqrt{\sum_{k_0=1}^{K_0} \sum_{k_2=1}^{K_2} \gamma_{jl}^{0,2} \, \mathrm{SW}_2^2(\mu_0^{k_0}, \mu_2^{k_2})} = \sqrt{\sum_{k_0=1}^{K_0} \sum_{k_1=1}^{K_1} \sum_{k_2=1}^{K_2} \eta_{k_0, k_1, k_2} \, \mathrm{SW}_2^2(\mu_0^{k_0}, \mu_2^{k_2})}$$

$$\leq \sqrt{\sum_{k_0=1}^{K_0} \sum_{k_1=1}^{K_1} \gamma_{k_0, k_1}^{0,1,*} \, \mathrm{SW}_2^2(\mu_0^{k_0}, \mu_1^{k_1})} + \sqrt{\sum_{k_1=1}^{K_1} \sum_{k_2=1}^{K_2} \gamma_{k_1, k_2}^{1,2,*} \, \mathrm{SW}_2^2(\mu_1^{k_1}, \mu_2^{k_2})}$$

$$= \mathrm{MSW}(\mu_0, \mu_1) + \mathrm{MSW}(\mu_1, \mu_2),$$

where we exploited the triangle inequality of $\mathrm{SW}_2$ and of the weighted 2-norm.

It remains to show the definiteness. If $\mu_0 = \mu_1$, then $\mathrm{MW}(\mu_0, \mu_1) = 0$, which is inherited to the other distances. Contrary, $\mathrm{DSMW}(\mu_0, \mu_1) = 0$ implies $\mathrm{SW}_2(\nu_\theta(\mu_0), \nu_\theta(\mu_1)) = 0$ for a.e. $\theta \in \mathbb{S}^{d-1}$. Since $\mathrm{SW}_2$ is a metric, we have $\nu_\theta(\mu_0) = \nu_\theta(\mu_1)$ and $\pi_{\theta, \sharp}(\mu_0) = \pi_{\theta, \sharp}(\mu_1)$ for a.e. $\theta$. Hence

$$0 = \int_{\mathbb{S}^{d-1}} \mathrm{W}_2(\pi_{\theta, \sharp} \mu_0, \pi_{\theta, \sharp} \mu_1) \, \mathrm{d}\theta = \mathrm{SW}_2(\mu_0, \mu_1)$$

implying $\mu_0 = \mu_1$. Due to (9), this is transferred to SMW and MSW.

$\square$

## 3.2 Strong and weak equivalences

We next study the topologies induced by the proposed sliced metrics. More precisely, we are interested in reverting the inequalities in Theorem 3.1 or, if this is not possible, in weak equivalences. For the majority of the results in this section, we assume that the means and covariances of the involved Gaussians are contained in a compact set. For a parameter set $P \subset \mathbb{R}^d \times \mathrm{S}_d^+$, we define

$$\mathrm{GMM}_{d,P}(K) := \{\mu \in \mathrm{GMM}_d(K) \mid (m^k, \Sigma^k) \in P\}.$$

This expression is again extended to $\mathrm{GMM}_{d,P}(\infty)$ by taking the union over all $K \in \mathbb{N}$. From the view of practical machine learning tasks, the compactness assumption is unproblematic as long as we consider GMMs whose variances and means are bounded for instance. In this case, $P$ can be chosen as sufficiently large closed ball in $\mathbb{R}^d \times \mathrm{S}_d^+$. For compact $P$, we can transfer the strong equivalence (3) to SMW and DSMW.

**Theorem 3.2.** SMW *and DSMW are 6-strongly equivalent on* $\mathrm{GMM}_{d,P}(\infty)$ *with* $P \subset \mathbb{R}^d \times \mathrm{S}_d^+$ *compact, i.e., there exists* $C_P > 0$ *such that*

$$\mathrm{DSMW}(\mu_0, \mu_1) \leq \mathrm{SMW}(\mu_0, \mu_1) \leq C_P \, \mathrm{DSMW}(\mu_0, \mu_1)^{\frac{1}{6}} \quad \forall \mu_0, \mu_1 \in \mathrm{GMM}_{d,P}(\infty).$$

*Proof.* For $\theta \in \mathbb{S}^{d-1}$, and $\mu := \sum_{k=1}^K \mu^k \in \mathrm{GMM}_d(K)$ with $\mu^k \sim \mathcal{N}(m^k, \Sigma^k)$, we observe $\pi_{\theta,\sharp}\mu \in \mathrm{GMM}_1(K)$, where the means and standard deviations of the projected components $\pi_{\theta,\sharp}\mu^k$ are given by $m_\theta^k := \theta \cdot m^k$ and $\sigma_\theta^k := (\theta^\top \Sigma^k \theta)^{\frac{1}{2}}$, i.e., the parameters are transformed by $T_\theta(m, \Sigma) := (\theta \cdot m, (\theta^\top \Sigma \theta)^{\frac{1}{2}})$. Due to the compactness of $P$, we find $X \subset \mathbb{R} \times \mathbb{R}_{\geq 0}$ compact, such that $T_\theta(P) \subset X$ for all $\theta \in \mathbb{S}^{d-1}$. Applying (3) pointwisely and Jensen's inequality, we have

$$\mathrm{SMW}^2(\mu_0, \mu_1) = \int_{\mathbb{S}^{d-1}} \mathrm{W}_2^2(\nu_\theta(\mu_0), \nu_\theta(\mu_1)) \, \mathrm{d}\theta$$

$$\leq C_X \int_{\mathbb{S}^{d-1}} \mathrm{SW}_2^{\frac{1}{3}}(\nu_\theta(\mu_0), \nu_\theta(\mu_1)) \, \mathrm{d}\theta \leq C_X \, \mathrm{DSMW}^{\frac{1}{3}}(\mu_0, \mu_1)$$

for all $\mu_i \in \mathrm{GMM}_d(\infty)$. $\qquad\square$

Generally, Wasserstein and sliced Wasserstein distances are not 1-strongly equivalent (Park & Slepčev, 2025). Similarly, no 1-strong equivalence (3) can be encountered for MW and MSW. Moreover, the counterexample in the following proof indicates that 1-strong equivalence may not even hold when restricting to Gaussians whose parameters are contained in compact sets.

**Proposition 3.3.** MW *and MSW are not 1-strongly equivalent on* $\mathrm{GMM}_d(K)$ *with* $d > 1$.

*Proof.* We restrict ourselves to the case $d = 2$ and follow a construction first presented in the errata of (Bayraktar & Guo, 2021). More precisely, we consider

$$\mu_\epsilon \sim \mathcal{N}\left(\left(\begin{smallmatrix} 0 \\ 0 \end{smallmatrix}\right), \left(\begin{smallmatrix} 1 & 0 \\ 0 & \epsilon \end{smallmatrix}\right)\right) \quad \text{and} \quad \mu \sim \mathcal{N}\left(\left(\begin{smallmatrix} 0 \\ 0 \end{smallmatrix}\right), \left(\begin{smallmatrix} 1 & 0 \\ 0 & 0 \end{smallmatrix}\right)\right).$$

Since $\mathrm{MW} \equiv \mathrm{W}_2$ and $\mathrm{MSW} \equiv \mathrm{SW}_2$ on $\mathrm{GMM}_2(1)$, the calculations in (Bayraktar & Guo, 2021) imply

$$\frac{\mathrm{MW}(\mu_\epsilon, \mu)}{\mathrm{MSW}(\mu_\epsilon, \mu)} = \frac{\mathrm{W}_2(\mu_\epsilon, \mu)}{\mathrm{SW}_2(\mu_\epsilon, \mu)}$$

$$= \frac{\epsilon}{(2 + \epsilon)\pi - 2 \int_0^{2\pi} |\cos^2(\phi)| (\cos^2(\phi) + \epsilon \sin^2(\phi))^{\frac{1}{2}} \, \mathrm{d}\phi} \to \infty.$$

Using L'Hôpital's rule and Lebesgue's dominated convergence theorem, we observe that the right-hand side diverges for $\epsilon \to 0$; so MW and MSW cannot be 1-strongly equivalent. The construction can be generalized to $d > 2$ by extending the diagonals by $\epsilon$ and 0 respectively. $\qquad\square$

For GMMs with parameters in a compact set, for instance, GMMs with bounded means and variances, we can establish the weak equivalence of the remaining distances. Intriguingly, the weak equivalence holds true although SMW is not immediately related to MSW and MW. While we conjecture that the weak equivalence holds true for all GMMs, the employed proof technique does not allow to skip the compactness assumption.

**Theorem 3.4.** DSMW, SMW, MSW, *and MW are weakly equivalent on* $\mathrm{GMM}_{d,P}(K)$ *with* $P \subset \mathbb{R}^d \times \mathrm{S}_d^+$ *compact, i.e.,*

$$\mathrm{DSMW}(\mu_n, \mu) \to 0 \quad \Leftrightarrow \quad \mathrm{SMW}(\mu_n, \mu) \to 0 \quad \Leftrightarrow \quad \mathrm{MSW}(\mu_n, \mu) \to 0 \quad \Leftrightarrow \quad \mathrm{MW}(\mu_n, \mu) \to 0$$

*as* $n \to \infty$ *for all* $(\mu_n)_{n \in \mathbb{N}} \subset \mathrm{GMM}_{d,P}(K)$ *and* $\mu \in \mathrm{GMM}_{d,P}(K)$.

To show this statement, we require the following lemma ensuring that almost all ($\forall\!\!\!\forall$) projections of two Gaussians do not collide.

**Lemma 3.5.** *For non-equal* $\mu_0, \mu_1 \in \mathrm{GMM}_d(1)$, *it holds*

$$\pi_{\theta,\sharp}\mu_0 \neq \pi_{\theta,\sharp}\mu_1 \quad \forall\!\!\!\forall \theta \in \mathbb{S}^{d-1}.$$

*Proof.* We consider non-equal $\mu_i \sim \mathcal{N}(m_i, \Sigma_i)$. If $m_0 \neq m_1$, then $\theta \cdot (m_0 - m_1) \neq 0$ for a.e. $\theta \in \mathbb{S}^{d-1}$, yielding the assertion. Otherwise, if $m_0 = m_1$ and $\Sigma_0 \neq \Sigma_1$, then $\theta^\top (\Sigma_0 - \Sigma_1) \theta = 0$ implies $\theta \in \ker(\Sigma_0 - \Sigma_1)$. Since $\dim(\ker(\Sigma_0 - \Sigma_1)) < d$, $\theta \notin \ker(\Sigma_0 - \Sigma_1) \cap \mathbb{S}^{d-1}$ almost surely, which finishes the proof. $\square$

*Proof of Theorem 3.4.* Let $\mu_n := \sum_{k_0=1}^K \omega_n^{k_0} \mu_n^{k_0}$ and $\mu := \sum_{k_1=1}^K \omega^{k_1} \mu^{k_1}$ in $\mathrm{GMM}_{d,P}(K)$ satisfy

$$\mathrm{SMW}^2(\mu_n, \mu) = \int_{\mathbb{S}^{d-1}} \sum_{k_0=1}^K \sum_{k_1=1}^K \gamma_{k_0,k_1}^{n,\theta,*} \mathrm{W}_2^2(\pi_{\theta,\sharp} \mu_n^{k_0}, \pi_{\theta,\sharp} \mu^{k_1}) \, \mathrm{d}\theta \to 0,$$

where $\gamma^{n,\theta,*}$ denotes the optimal MW plan for fixed $\theta \in \mathbb{S}^{d-1}$. Due to the $L^2(\mathbb{S}^{d-1})$ convergence, for any subsequence of $\mu_n$, we find a further subsequence $(\mu_{n_j})_{j \in \mathbb{N}}$ such that

$$\sum_{k_0=1}^K \sum_{k_1=1}^K \gamma_{k_0,k_1}^{n_j,\theta,*} \mathrm{W}_2^2(\pi_{\theta,\sharp} \mu_{n_j}^{k_0}, \pi_{\theta,\sharp} \mu^{k_1}) \to 0 \quad \forall\forall\theta \in \mathbb{S}^{d-1} \tag{10}$$

pointwisely. Since $P$ is compact, $(\mu_{n_j})_{j \in \mathbb{N}}$ can be chosen such that $\mu_{n_j}^{k_0}$ converges to some $\tilde{\mu}^{k_0} \in \mathrm{GMM}_d(1)$ and $\omega_{n_j}^{k_0}$ to some $\tilde{\omega}^{k_0} \in \mathbb{R}_{\geq 0}$.

Assume that there exists $\tilde{\mu}^{k_0}$ with $\tilde{\omega}^{k_0} > 0$ such that $\tilde{\mu}^{k_0} \neq \mu^{k_1}$ for all $k_1$ with $\omega^{k_1} > 0$. Due to Lemma 3.5, we find $\theta \in \mathbb{S}^{d-1}$ such that (10) converges and

$$\pi_{\theta,\sharp} \tilde{\mu}^{k_0} \neq \pi_{\theta,\sharp} \mu^{k_1} \quad \forall k_1 \in \{k \mid \omega^k > 0\}.$$

Since $\mathrm{W}_2^2(\pi_{\theta,\sharp} \mu_{n_j}^{k_0}, \pi_{\theta,\sharp} \mu^{k_1})$ is thus bounded away from zero for $n_j$ large enough, the pointwise convergence in (10) implies $\gamma_{k_0,k_1}^{n_j,\theta,*} \to 0$ for all $k_1 \in \{k \mid \omega^k > 0\}$, i.e., an entire row in $\gamma^{n_j,\theta,*}$ which sums up to a non-zero weight vanishes. This, however, contradicts that $\gamma^{n_j,\theta,*}$ is a discrete transport plan. Consequently, $\mu_{n_j}^{k_0}$ converges to a non-zero component of $\mu$.

For any $\theta \in \mathbb{S}^{d-1}$ such that (10) hold true and $\pi_{\theta,\sharp} \tilde{\mu}^{k_0} \neq \pi_{\theta,\sharp} \mu^{k_1}$ for the components with non-zero weights, we observe

$$\gamma_{k_0,k_1}^{n_j,\theta,*} \to 0 \quad \begin{cases} \text{if } \tilde{\mu}^{k_0} \neq \mu^{k_1} \text{ and } \tilde{\omega}^{k_0}, \omega^{k_1} > 0, \\ \text{or if } \tilde{\omega}^{k_0} = 0, \end{cases}$$

and hence

$$\gamma_{k_0,k_1}^{n_j,\theta,*} \to \tilde{\omega}^{k_0} \quad \text{if } \tilde{\mu}^{k_0} = \mu^{k_1} \text{ and } \tilde{\omega}^{k_0}, \omega^{k_1} > 0.$$

Exploiting that the Wasserstein distances between all involved Gaussians in $\mathrm{GMM}_{d,P}(1)$ are bounded, we finally have

$$\mathrm{MW}^2(\mu_n, \mu) \leq \sum_{k_0=1}^K \sum_{k_1=1}^K \gamma_{k_0,k_1}^{n_j,\theta,*} \mathrm{W}_2^2(\mu_{n_j}^{k_0}, \mu^{k_1}) \to 0 \quad \text{as} \quad j \to \infty.$$

Because this holds true for any subsequence of $(\mu_n)_{n \in \mathbb{N}}$, the assertion is established. The remaining implications follow from the inequalities in Theorem 3.1 and 3.2. $\square$

### 3.3 Weak topology

Beyond computational speed-up, the sliced Wasserstein distance is closely related to the original Wasserstein distance. Specifically, both imply weak convergence and thus induce the weak topology on $\mathcal{P}_2(\mathbb{R}^d)$, see (Nadjahi et al., 2020). In this section, we aim to explore this property for our proposed sliced distances. In a first step, we establish an analogue of (8) for SMW.

**Proposition 3.6.** *For $\mu_i \in \mathrm{GMM}_d(K_i)$, MSW satisfies*

$$\mathrm{SW}_2(\mu_0, \mu_1) \leq \mathrm{SMW}(\mu_0, \mu_1)$$

$$\leq \mathrm{SW}_2(\mu_0, \mu_1) + \sqrt{2 \sum_{k_0=1}^{K_0} \omega_0^{k_0} \int_{\mathbb{S}^{d-1}} \theta^\top \Sigma_0^{k_0} \theta \, \mathrm{d}\theta} + \sqrt{2 \sum_{k_1=1}^{K_1} \omega_1^{k_1} \int_{\mathbb{S}^{d-1}} \theta^\top \Sigma_1^{k_1} \theta \, \mathrm{d}\theta}.$$

*Proof.* The lower bound follows from the definition of MSW and $\mathrm{W}_2(\mu_0, \mu_1) \leq \mathrm{MW}(\mu_0, \mu_1)$. The upper bound follows from applying (8) in the definition of MSW together with the triangle inequality on $L^2(\mathbb{S}^{d-1})$.

$\qquad\square$

Since $\mathrm{SW}_2$ induces the weak topology, also MSW yields the weak convergence. Delon and Desolneux (Delon & Desolneux, 2020) give an example that MW and $\mathrm{W}_2$ are not weakly equivalent and that the upper bound in (8) is tight. This example carries over to $\mathrm{SW}_2$ and SMW. Owing to the additional slicing in DSMW, which decreases SMW, Proposition 3.6 cannot be exploited to study the relation of DSMW to the weak topology. For GMMs whose parameters are contained in a compact set, the following statement yields a first approach in this direction.

**Proposition 3.7.** *For $P \subset \mathbb{R}^d \times \mathrm{S}_d^+$ compact, there exists $C_p > 0$ such that*

$$\mathrm{SW}_2^6(\mu_0, \mu_1) \leq C_P \, \mathrm{DSMW}(\mu_0, \mu_1) \quad \forall \mu_0, \mu_1 \in \mathrm{GMM}_{d,P}(\infty).$$

*Proof.* The statement immediately follows from combining the strong equivalence in Theorem 3.2 with the lower bound in Proposition 3.6.

$\qquad\square$

As $\mathrm{SW}_2$ metricizes weak convergence, $\lim_{n\to\infty} \mathrm{DSMW}(\mu_n, \mu) = 0$ results in $\mu_n \rightharpoonup \mu$ on $\mathrm{GMM}_{d,P}(\infty)$. Proposition 3.7 does not imply the strong equivalence of $\mathrm{SW}_2$ and DSMW on $\mathrm{GMM}_{d,P}(\infty)$ since the second bound is not established. Beyond compact parameter sets, $\mathrm{SW}_2$ and DSMW are not even weakly equivalent. Henceforth, $\mathbf{0}_d \in \mathbb{R}^d$ denotes the all-zero vector, $\boldsymbol{I}_d \in \mathbb{R}^{d\times d}$ the identity matrix, and $\Gamma$ the gamma function.

**Proposition 3.8.** *For $m_{k_0} \in \mathbb{R}^d$ and $\sigma > 0$, let $\mu_0 = \sum_{k_0=1}^{K_0} \frac{1}{K_0} \delta_{m_{k_0}}$ and $\mu_1 \sim \mathcal{N}(\mathbf{0}_d, \sigma^2 \boldsymbol{I}_d)$ be given. Then*

$$\mathrm{DSMW}^2(\mu_0, \mu_1) \geq \frac{2\sigma^2 \pi^{\frac{d+2}{2}}}{\Gamma(\frac{d}{2})}.$$

*Proof.* Due to the uniqueness of the transport plan between an one-point measure and any other measure, a direct calculation yields

$$\begin{aligned}
\mathrm{DSMW}^2(\mu_0, \mu_1) &= \int_{\mathbb{S}^{d-1}} \mathrm{SW}_2^2(\nu_\theta(\mu_0), \nu_\theta(\mu_1)) \, \mathrm{d}\theta \\
&= \int_{\mathbb{S}^{d-1}} \int_0^{2\pi} \frac{1}{K_0} \sum_{k_0=1}^{K_0} \big( (\theta \cdot m_{k_0}) \cos(\phi) + \sigma \sin(\phi) \big)^2 \, \mathrm{d}\phi \, \mathrm{d}\theta \\
&= \int_{\mathbb{S}^{d-1}} \int_0^{2\pi} \frac{1}{K_0} \sum_{k_0=1}^{K_0} \big[ (\theta \cdot m_{k_0})^2 \cos^2(\phi) + \sigma^2 \sin^2(\phi) \big] \, \mathrm{d}\phi \, \mathrm{d}\theta.
\end{aligned}$$

Here, the mixed terms in the extensions of the squares vanish since they are integrals over odd functions. Dropping the cosine terms, we obtain

$$\mathrm{DSMW}^2(\mu_0, \mu_1) \geq \frac{2\sigma^2 \pi^{\frac{d}{2}}}{\Gamma(\frac{d}{2})} \int_0^{2\pi} \sin^2(\phi) \, \mathrm{d}\phi = \frac{2\sigma^2 \pi^{\frac{d+2}{2}}}{\Gamma(\frac{d}{2})}. \qquad\square$$

For a sequence of point measures $\mu_n$ approximating a Gaussian $\mathcal{N}(\mathbf{0}_d, \sigma^2 \boldsymbol{I}_d)$, i.e., $\lim_{n\to\infty} \mathrm{SW}_2(\mu_n, \mu) = 0$, Proposition 3.8 implies $\lim_{n\to\infty} \mathrm{DSMW}(\mu_n, \mu) \geq 2\sigma^2 \pi^{\frac{d+2}{2}}/\Gamma(\frac{d}{2}) \neq 0$; so $\mathrm{SW}_2$ and DSMW cannot be weakly equivalent.

In particular, this means that samples from a Gaussian do not yield a (weak) approximation.

---

**Algorithm 1** Implementation of MSW

---

**Require:** GMMs $\mu_i := \sum_{k_i=1}^{K_i} \omega_i^{k_i} \mu_i^{k_i}$ with $\mu_i^{k_i} \sim \mathcal{N}(m_i^{k_i}, \Sigma_i^{k_i})$ ▷ $i := 0, 1$
**Require:** number of projections $L$
1: **for** $k_i := 1, \dots, K_i,\ i := 0, 1$ **do**

2: $\quad c_{k_0, k_1} := \dfrac{1}{L} \sum_{\ell=1}^{L} (\theta_\ell \cdot m_0^{k_0} - \theta_\ell \cdot m_1^{k_1})^2 + \left( (\theta_\ell^\top \Sigma_0^{k_0} \theta_\ell)^{\frac{1}{2}} - (\theta_\ell^\top \Sigma_1^{k_1} \theta_\ell)^{\frac{1}{2}} \right)^2$

$\qquad\qquad\qquad\qquad\qquad\qquad\qquad\qquad\qquad\qquad\qquad\qquad\qquad\qquad$ ▷ $\theta_\ell \leftarrow \mathcal{U}(\mathbb{S}^{d-1})$

3: **end for**
4: **return** $\min_{\gamma \in \Gamma(\omega_0, \omega_1)} \sum_{k_0=1}^{K_0} \sum_{k_1=1}^{K_1} \gamma_{k_0, k_1}\, c_{k_0, k_1}$ ▷ optimal transport solver

---

## 4 Numerical analysis

In this section, we present the computational implementation of MSW and DSMW, a runtime comparison with MW, and numerical experiments showcasing the potential of our novel sliced metrics. Experiments were conducted on a system equipped with a 13th Gen Intel Core i5-13600K CPU and an NVIDIA GeForce RTX 3060 GPU with 12 GB of memory. Our implementation[1] is based on Python 3.12 and employs the Python optimal transport library (POT) (Flamary et al., 2021) for optimal transport solvers and PyTorch (Paszke et al., 2019) for automatic differentiation.

### 4.1 Implementation details

A common, popular approach to implement the sliced Wasserstein distance relies on the so-called Monte Carlo integration (Nadjahi et al., 2020). For this, we independently draw $L$ random samples $\theta_\ell \in \mathbb{S}^{d-1}$ with respect to the uniform distribution $\mathcal{U}(\mathbb{S}^{d-1})$ on the sphere (in pseudocode: $\theta_\ell \leftarrow \mathcal{U}(\mathbb{S}^{d-1})$) and approximate the sliced $p$-Wasserstein distance by

$$\widehat{\mathrm{SW}}_{p,L}(\mu_0, \mu_1) := \left( \frac{1}{L} \sum_{\ell=1}^{L} \mathrm{W}_p^p(\pi_{\theta_\ell, \sharp}\, \mu_0, \pi_{\theta_\ell, \sharp}\, \mu_1) \right)^{\frac{1}{p}}, \tag{11}$$

where the Wasserstein distances on the line can, in general, be calculated via (2). Independent of the dimension $d$, the Monte Carlo approximation converges with rate $\mathcal{O}(L^{-\frac{1}{2}})$. More precisely, denoting the uniform measure by $u$, we have the following estimate.

**Theorem 4.1 (**Nadjahi et al., 2020, Thm 6)**.** *For $\mu_0, \mu_1 \in \mathcal{P}_p(\mathbb{R}^d)$, and independent samples $\theta_\ell$, $\ell = 1, \dots, L$, with respect to $\mathcal{U}(\mathbb{S}^{d-1})$, the expected absolute error is bounded by*

$$\mathbb{E}_{\theta_1, \dots, \theta_L} |\widehat{\mathrm{SW}}_{p,L}^p(\mu_0, \mu_1) - \mathrm{SW}_p^p(\mu_0, \mu_1)|$$
$$\leq \frac{1}{\sqrt{L}} \left( \int_{\mathbb{S}^{d-1}} |\mathrm{W}_p^p(\pi_{\theta, \sharp}\, \mu_0, \pi_{\theta, \sharp}\, \mu_1) - \mathrm{SW}_p^p(\mu_0, \mu_1)|^2\, \mathrm{d}u(\theta) \right)^{\frac{1}{2}}.$$

The term on the right-hand side may be interpreted as the standard deviation of the Wasserstein distance $\mathrm{W}_p^p(\pi_{\theta, \sharp}\, \mu_0, \pi_{\theta, \sharp}\, \mu_1)$ on the line with respect to $\mathcal{U}(\mathbb{S}^{d-1})$. Incorporating the slicing of Gaussians and the 1d Wasserstein distance (5), we obtain a simple closed-form expression for the outer costs of MSW. The corresponding transport problem may then be solved using POT (Flamary et al., 2021). The details of the resulting MSW implementation are presented in Algorithm 1. The Monte Carlo integration may also be replaced by quasi-Monte Carlo methods (Nguyen et al., 2024; Hertrich et al., 2025).

For the double slicing behind DSMW, we have to implement the following two major steps: 1. Determine the parameters of the GMMs $\pi_{\theta, \sharp}\, \mu_i$, i.e., calculate $\nu_\theta(\mu_i) \in \mathcal{P}_2(\mathbb{R} \times \mathbb{R}_{\geq 0})$. This step can be realized by transforming the parameters of the Gaussian components via $(m, \Sigma) \mapsto (\theta \cdot m, (\theta^\top \Sigma \theta)^{\frac{1}{2}})$. 2. Slice the resulting point measure in $\mathcal{P}_2(\mathbb{R} \times \mathbb{R}_{\geq 0})$. Using the standard parametrization of $\mathbb{S}^1$, we realize this step by

---

[1] https://github.com/MoePien/sliced_OT_for_GMMs

---

**Algorithm 2** Implementation of DSMW

---

**Require:** GMMs $\mu_i := \sum_{k_i=1}^{K_i} \omega_i^{k_i} \mu_i^{k_i}$ with $\mu_i^{k_i} \sim \mathcal{N}(m_i^{k_i}, \Sigma_i^{k_i})$ $\qquad\qquad\qquad\qquad\qquad \triangleright i := 0, 1$
**Require:** number of projections $L$
1: **for** $\ell := 1, \ldots, L$ **do**
2: $\quad \theta_\ell \leftarrow \mathcal{U}(\mathbb{S}^{d-1}), \ \phi_\ell \leftarrow \mathcal{U}([0, 2\pi))$
3: $\quad \xi_{i,\ell} := \sum_{k_i=1}^{K_i} \omega_i^{k_i} \delta_{p_{i,\ell}^{k_i}}, \quad p_{i,\ell}^{k_i} := (\theta_\ell \cdot m_i^{k_i})\cos(\phi_\ell) + (\theta_\ell^\top \Sigma_i^{k_i} \theta_\ell)^{\frac{1}{2}} \sin(\phi_\ell)$
$\qquad\qquad\qquad\qquad\qquad\qquad\qquad\qquad\qquad\qquad\qquad\qquad\qquad\qquad\qquad\qquad \triangleright i := 0, 1$
4: $\quad w_\ell := \mathrm{W}_2^2(\xi_{0,\ell}, \xi_{1,\ell})$ $\qquad\qquad\qquad\qquad\qquad\qquad\qquad \triangleright$ 1d optimal transport via (2)
5: **end for**
6: **return** $\frac{1}{L} \sum_{\ell=1}^{L} w_\ell$

---

calculating the inner product with $(\cos\phi, \sin\phi)$ where $\phi \in [0, 2\pi)$. Concatenation of both steps yields the parameter transformation $\xi_{\theta,\phi} \colon \mathrm{GMM}(\infty) \to \mathcal{P}_2(\mathbb{R})$ that is, for fixed $\theta \in \mathbb{S}^{d-1}$ and $\phi \in [0, 2\pi)$, given by

$$\xi_{\theta,\phi}(\mu) := \sum_{k=1}^{K} \omega^k \delta_{p^k}, \quad p^k := (\theta \cdot m_k)\cos(\phi) + (\theta^\top \Sigma^k \theta)^{\frac{1}{2}} \sin(\phi).$$

Drawing $L$ independent samples $(\theta_\ell, \phi_\ell)$ with respect to the uniform measure $\mathcal{U}(\mathbb{S}^{d-1} \times [0, 2\pi))$, we propose to approximate DSMW by

$$\widehat{\mathrm{DSMW}}_L(\mu_0, \mu_1) := \Big(\frac{1}{L} \sum_{\ell=1}^{L} \mathrm{W}_2^2(\xi_{\theta_\ell,\phi_\ell}(\mu_0), \xi_{\theta_\ell,\phi_\ell}(\mu_1))\Big)^{\frac{1}{2}}. \tag{12}$$

The details of the implementation are given in Algorithm 2, where we again rely on POT (Flamary et al., 2021) to compute the 1d optimal transports. Similarly to $\mathrm{SW}_p$, the simultaneous Monte Carlo integration with respect to both slicing directions yields a convergence rate of $\mathcal{O}(L^{-\frac{1}{2}})$.

**Theorem 4.2.** *For $\mu_0, \mu_1 \in \mathrm{GMM}_d(\infty)$, and independent samples $(\theta_\ell, \phi_\ell)$, $\ell = 1, \ldots, L$, with respect to $\mathcal{U}(\mathbb{S}^{d-1} \times [0, 2\pi))$, the expected absolute error is bounded by*

$$\mathbb{E}_{(\theta_1,\phi_1),\ldots,(\theta_L,\phi_L)} |\widehat{\mathrm{DSMW}}_L^2(\mu_0, \mu_1) - \mathrm{DSMW}^2(\mu_0, \mu_1)|$$
$$\leq \frac{1}{\sqrt{L}} \Big(\int_{\mathbb{S}^{d-1} \times [0,2\pi]} |\mathrm{W}_2^2(\xi_{\theta,\phi}(\mu_0), \xi_{\theta,\phi}(\mu_1)) - \mathrm{DSMW}^2(\mu_0, \mu_1)|^2 \, \mathrm{d}u(\theta, \phi)\Big)^{\frac{1}{2}}.$$

*Proof.* Applying Hölder's inequality, and exploiting that the samples $(\theta_\ell, \phi_\ell)$ are drawn independently, we obtain

$$\mathbb{E}_{(\theta_1,\phi_1),\ldots,(\theta_L,\phi_L)} |\widehat{\mathrm{DSMW}}_L^2(\mu_0, \mu_1) - \mathrm{DSMW}^2(\mu_0, \mu_1)|$$
$$= \int \cdots \int_{(\mathbb{S}^{d-1} \times [0,2\pi])^L} |\widehat{\mathrm{DSMW}}_L^2(\mu_0, \mu_1) - \mathrm{DSMW}^2(\mu_0, \mu_1)| \, \mathrm{d}u(\theta_1, \phi_1) \cdots \mathrm{d}u(\theta_L, \phi_L)$$
$$\leq \Big(\int \cdots \int_{(\mathbb{S}^{d-1} \times [0,2\pi])^L} |\widehat{\mathrm{DSMW}}_L^2(\mu_0, \mu_1) - \mathrm{DSMW}^2(\mu_0, \mu_1)|^2 \, \mathrm{d}u(\theta_1, \phi_1) \cdots \mathrm{d}u(\theta_L, \phi_L)\Big)^{\frac{1}{2}}$$
$$= \Big(\frac{1}{L} \sum_{\ell=1}^{L} \int_{\mathbb{S}^{d-1} \times [0,2\pi]} |\mathrm{W}_2^2(\xi_{\theta_\ell,\phi_\ell}(\mu_0), \xi_{\theta_\ell,\phi_\ell}(\mu_1)) - \mathrm{DSMW}^2(\mu_0, \mu_1)|^2 \, \mathrm{d}u(\theta_\ell, \phi_\ell)\Big)^{\frac{1}{2}}$$
$$= \frac{1}{\sqrt{L}} \Big(\int_{\mathbb{S}^{d-1} \times [0,2\pi]} |\mathrm{W}_2^2(\xi_{\theta,\phi}(\mu_0), \xi_{\theta,\phi}(\mu_1)) - \mathrm{DSMW}^2(\mu_0, \mu_1)|^2 \, \mathrm{d}u(\theta, \phi)\Big)^{\frac{1}{2}}. \qquad \square$$

Similarly to the convergence rate for $\mathrm{SW}_p$, the integral on the right-hand side corresponds to the variance of $\mathrm{W}_2^2(\xi_{\theta,\phi}(\mu_0), \xi_{\theta,\phi}(\mu_1))$ with respect to $\mathcal{U}(\mathbb{S}^{d-1} \times [0, 2\pi))$.

Table 1: Average CPU computation time in seconds (mean plus-minus standard deviation) for GMMs with varying dimensions $d$ and component number $K$ for MW, MSW, and DSMW. Additionally, we approximate the outer transport behind MW using efficient Sinkhorn iterations with regularization parameter $\epsilon = 0.1$ (entropic MW). The time for MW quickly surpasses 60 seconds, whereas MSW takes only seconds and DSMW is calculated within a single second. The Monte Carlo iterations are stopped when reaching a precision of $10^{-3}$. Note that GPU usage achieves further acceleration in practice.

| dim $d$ | components $K$ | | | | dim $d$ | components $K$ | | |
|---|---|---|---|---|---|---|---|---|
| | 10 | 100 | 500 | | | 10 | 100 | 500 |
| 10 | 0.05±0.08 | 1.85±0.02 | 46.4±0.32 | | 10 | 0.02±0.00 | 1.67±0.02 | 41.57±0.27 |
| 100 | 2.59±0.19 | >60 | >60 | | 100 | 2.74±0.35 | >60 | >60 |
| 500 | >60 | >60 | >60 | | 500 | >60 | >60 | >60 |
| | MW | | | | | Entropic MW | | |

| dim $d$ | components $K$ | | | | dim $d$ | components $K$ | | |
|---|---|---|---|---|---|---|---|---|
| | 10 | 100 | 500 | | | 10 | 100 | 500 |
| 10 | 1.21±0.15 | 1.40±0.19 | 1.28±0.09 | | 10 | 0.15±0.05 | 0.02±0.01 | 0.01±0.01 |
| 100 | 0.89±0.42 | 3.11±0.15 | 10.2±0.92 | | 100 | 0.13±0.08 | 0.05±0.03 | 0.08±0.04 |
| 500 | 4.63±0.51 | 44.7±1.95 | >60 | | 500 | 0.34±0.24 | 0.89±0.41 | 1.05±0.48 |
| | MSW | | | | | DSMW | | |

## 4.2 Runtime comparison

To evaluate the computational speed-up of Algorithm 1 and 2, we present a side-by-side CPU time comparison in Table 1. For each dimension $d$ and component number $K$, we calculate MW, MSW, and DSMW for 10 pairs of random GMMs. The GMMs are generated with uniformly distributed weights in $\Delta_K$ and uniform means in $[0, 10]^d$. Based on the Cholesky decomposition, the covariance matrices are generated as $QQ^\top$ based on lower triangular matrices $Q$ with uniform entries in $[0, 1]$. The Monte Carlo estimates of $SW_2$ (11) for the MSW costs in Algorithm 1 and of DSMW (12) (Algorithm 2) are calculated up to a convergence. More precisely, we stop the Monte Carlo iteration as soon as the mean precision attains

$$\sum_{k_0=1}^{K_0} \sum_{k_1=1}^{K_1} \frac{|\widehat{SW}_{2,L}(\mu_0^{k_0}, \mu_1^{k_1}) - \widehat{SW}_{2,L-10}(\mu_0^{k_0}, \mu_1^{k_1})|}{K_0 K_1} \leq 10^{-3} \tag{13}$$

and

$$|\widehat{DSMW}_L(\mu_0, \mu_1) - \widehat{DSMW}_{L-10}(\mu_0, \mu_1)| \leq 10^{-3}; \tag{14}$$

so we stop as soon as the estimate nearly stagnates over 10 iterations. The number of 10 iterations is here chosen manually to reduce the stochastic impact of the random process on subsequent iterates. Table 1 shows that MW, unlike MSW and DSMW, does not scale well beyond toy GMMs. To reduce the numerical burden of the outer transport problem behind MW, we additionally solve this via efficient Sinkhorn iterations (Peyré & Cuturi, 2019) that rely on an entropy regularization. This adaption only yields a minor acceleration showing that the analytical solution of the inner transports are the major bottleneck. We observed an analogous slight acceleration for a regularized version of MSW—not recorded in Table 1. The usage of the original MW distance is thus limited in image reconstruction methods typically employing GMMs with 100–300 components on dimension of size 50–100 (Zoran & Weiss, 2011). In such cases, MSW and DSMW are computed in a second or less, whereas MW takes minutes. Interestingly, we consistently observed accelerated convergence of DSMW for higher dimensions for $K < 500$.

To study the behavior of MSW and DSMW for GMMs with higher numbers of components, we perform additional experiments in dimension $d = 50$, see Table 2. Moreover, we report the averaged number of

| | | components $K$ | | | |
|---|---|---|---|---|---|
| | | 1000 | 2000 | 5000 | 10000 |
| MSW | time | 8.9±1.03 | 47.13±2.81 | >60 | >60 |
| | proj. | 8967 | 8816 | — | — |
| DSMW | time | 0.02±0.01 | 0.04±0.01 | 0.09±0.03 | 0.16±0.0 |
| | proj. | 15 | 15 | 12 | 10 |

Table 2: Average CPU computation time in seconds (mean plus-minus standard deviation) and average number of projections until convergence for GMMs with varying component numbers $K$ in dimension $d = 50$.

| | MSW | | DSMW | |
|---|---|---|---|---|
| dim $d$ | time | proj. | time | proj. |
| 1000 | 15.40±2.20 | 6932 | 1.00±0.47 | 225 |
| 2000 | 53.09±7.82 | 6928 | 3.43±1.72 | 228 |
| 3000 | >60 | — | 6.90±3.21 | 215 |

Table 3: Average CPU computation time in seconds (mean plus-minus standard deviation) and average number of projections until convergence for GMMs in varying dimensions $d$ and $K = 10$ components.

projections required to reach the convergence criteria (13) and (14). While the outer transport behind MSW leads to exploding computation times, DSMW is still computed in under one second. The averaged number of required projections slightly deceases for higher component numbers. The reason for this behaviour might be that more and more components in the generated GMMs become alike. For very high component numbers, the entire generated GMMs are closely related such that the variance in the upper bound from Theorem 4.2 becomes small; so the absolute error for DSMW decreases extremely rapid. This effect is much less distinct for MSW since the majority of the single components still differ.

Finally, we repeat the experiment for GMMs in higher dimensions, where the number of components is fixed by $K = 10$, see Table 3. As before, we observe a significant impact of the double slicing on the computation time. While MSW and DSMW take longer for increasing dimensions, the average projection number remains stable; so the increasing computation times mainly trace back to the increased numerical effort for each projection. Again, we observe that DSMW requires much less directions than MSW for convergence, which leads to an additional speed-up.

## 5 Experiments and applications

### 5.1 Detection of cluster number

When clustering or classifying data, a common approach is to fit a likelihood-maximizing GMM using the EM algorithm (Wan et al., 2019). For this, however, the number of Gaussian components has to be specified, which requires additional data analysis. In practice, we may rely on a wide range of classification indices and criteria to find a suitable number (Xu et al., 2016). As an alternative, we propose to use GMM-based metrics by tracking the distance between the fitted GMMs $\mu_k \in \text{GMM}_d(k)$ and $\mu_{k+1} \in \text{GMM}_d(k+1)$. Figuratively, we try to find the smallest $k \in \mathbb{N}$ such that adding more components does not change the fitted model. In other words, we increase $k$ until the distance between $\mu_k$ and $\mu_{k+1}$ vanishes. To illustrate this approach, we sample 1000 data points from four 2d distributions, each with four or six clearly separated modes. Then, GMMs $\mu_k$ with $k = 2, \ldots, 9$ Gaussian components are fitted using the EM algorithm, and the distances $\text{MW}(\mu_k, \mu_{k+1})$, $\text{MSW}(\mu_k, \mu_{k+1})$ and $\text{DSMW}(\mu_k, \mu_{k+1})$ are computed. The evolutions of the fitted models are presented in Figure 2. For each distribution, we observe convergence in all three metrics at $k = 4$ or $k = 6$, which corresponds to the true number of components. The experiment suggests that the GMM-based matrices can indeed be used to specify the number of clusters. A broader study and comparison with other methods, especially when the classes are not well separated, is left for future research.

### 5.2 Perceptual metric

The Fréchet Inception Distance (FID) (Heusel et al., 2017) is a widely used metric for evaluating generative models in computer vision. It estimates the distance between real and generated image distributions by

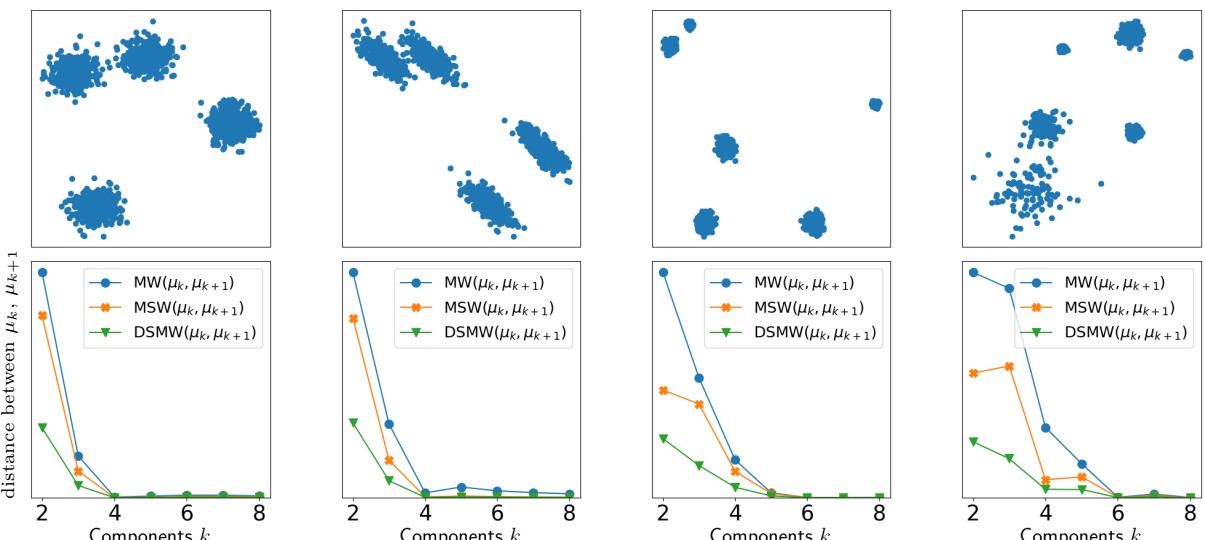

Figure 2: Detection of cluster numbers by tracking the distances between $\mu_k$ and $\mu_{k+1}$, where $\mu_k$ is a fitted GMM with $k$ components. For the input data (top), the plots (bottom, y-axis from 0 to maximum) show the distance between $\mu_k$ and $\mu_{k+1}$ with respect to MW (blue), MSW (orange), and DSMW (green). The distances become zero after $k = 4$ (left) and $k = 6$ (right) indicating that higher component numbers only yield reparametrizations of the former models. The first zero hence corresponds to the correct cluster number. Notice that MW seems to be slightly less stable than MSW and DSMW.

assuming that their deep feature representations follow multivariate Gaussian distributions. In more detail, the features of the given empirical image distributions are extracted using a pre-trained neural network. Both feature sets are then modeled as normal distributions, whose Wasserstein distance is estimated by (4). Despite good performance, the Gaussian assumption does not align with empirical feature distributions (Luzi et al., 2023). As an alternative, the Wasserstein on Mixtures (WaM) metric fits two GMMs to the feature distribution using the EM algorithm and estimates the mixture Wasserstein distance (7) to capture more visual information (Luzi et al., 2023). Both perceptual metrics—FID and WaM—have been shown to align with human perception.

In this experiment, we adapt the WaM metric by replacing MW with MSW and DSMW. As feature extractor for FID and WaM, we employ a pre-trained Inception-v3 network (Szegedy et al., 2016) with a latent representation of 2048 dimensions. The obtained feature set is then fitted by a multi-variate normal distribution or by a five-component GMM respectively. For the real image set, we employ a subset of 1000 CIFAR10 images (Krizhevsky, 2009). For the artificial set, we disturb these images with Gaussian noise (standard deviation between 0.01 and 0.2), Salt&Pepper noise (corruption ratio between 5% and 30%), and Gaussian blur (standard deviation between 0.1 and 1.5). The resulting FID and WaM between the real and artificial image sets are visualized in Figure 3. All perceptual metrics react very similarly to the different distortions. Stronger distortions lead to increased distances, and the FID and the (adapted) WaM curves display comparable shapes for the same distortion types. Given the fitted GMMs, MW calculation took around 11 minutes, MSW calculation took around 1 minute and DSMW calculation took around 20 seconds.

## 5.3 Quantization of Gaussian mixtures

The focus of the next experiment is to demonstrate that our DSMW distance may be used in the context of gradient-based optimization. Exemplarily, we aim to reduce the component number of a given GMM while preserving key statistical properties, aiding in model compression. Earlier work on this topic has explored optimal component selections and divergence-based approximations to minimize the information loss (Runnalls, 2007; Crouse et al., 2011; Assa & Plataniotis, 2018). To quantize a given $\mu \in \mathrm{GMM}_d(K)$, we

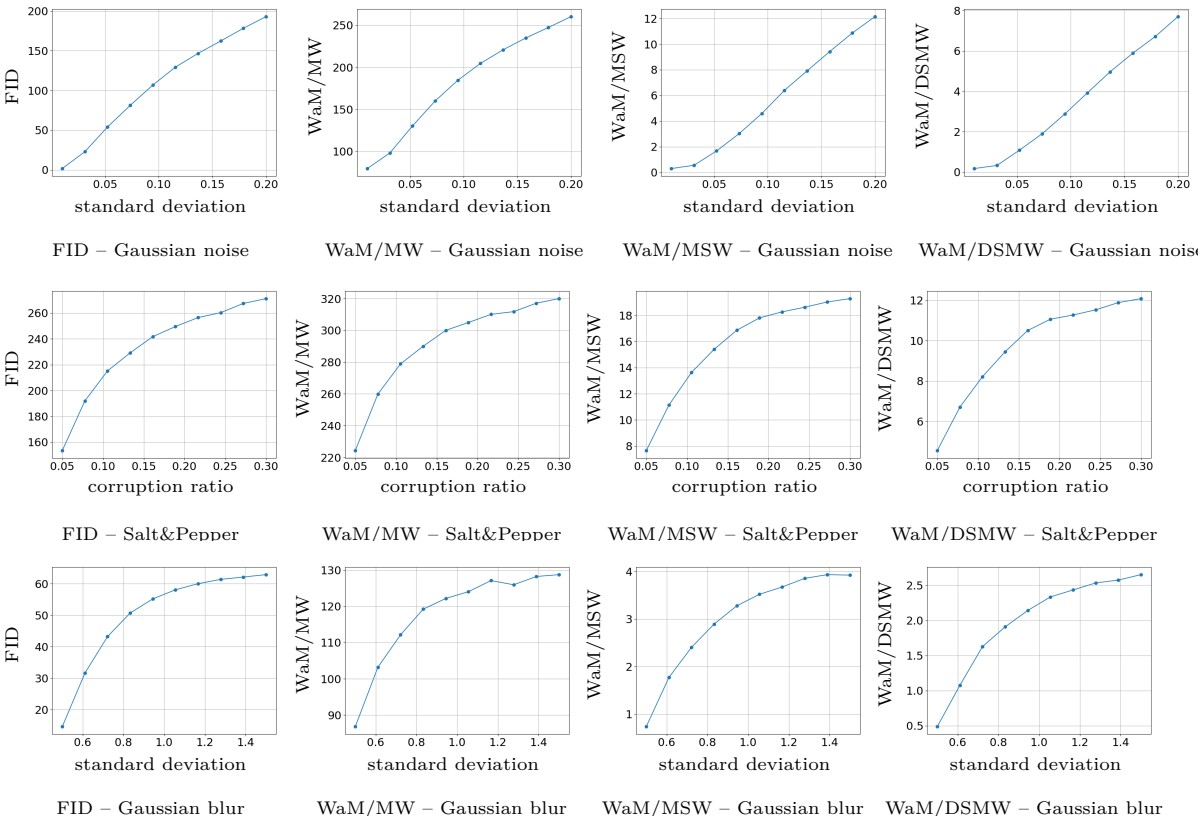

Figure 3: FID (left), WaM with MSW (center left), WaM with MW (center right), and WaM with DSMW (right) between real and artificial CIFAR10 image sets based on different types and levels of image distortions (standard deviation of Gaussian noise, corruption ratio of Salt&Pepper noise, standard deviation of Gaussian blur). All resulting curves of the evaluated perceptual metrics show comparable behaviors.

instead propose to solve

$$\min_{\mu^* \in \mathrm{GMM}_d(K^*)} \mathrm{DSMW}^2(\mu^*, \mu) \quad \text{with} \quad K^* < K.$$

In favor of a gradient descent minimization, we parametrize $\mu^*$ according to (Gepperth & Pfülb, 2021). For this, we denote the set of lower triangular matrices with non-negative diagonal by $\mathrm{Tri}_{\geq 0}(d)$. Given a parameter vector $\rho := (w^k, m^k, Q^k)_{k=1}^{K^*}$ with $w^k \in \mathbb{R}$, $m^k \in \mathbb{R}^d$, and $Q^k \in \mathrm{Tri}_{\geq 0}(d)$, we consider the parametrized GMM:

$$\mu_\rho := \sum_{k=1}^{K^*} \frac{\exp(w^k)}{\sum_{\ell=1}^{K^*} \exp(w^\ell)} \mu_\rho^k, \qquad \mu_\rho^k \sim \mathcal{N}(m_k, Q^k Q^{k,\top} + \sigma^2 \boldsymbol{I}_d), \tag{15}$$

where we add the identity $\boldsymbol{I}_d$ for numerical stability. Using a fixed number of random directions $\theta_\ell$ in (12), we employ the Adam scheme (Kingma & Ba, 2014) in combination with automatic differentiation to minimize

$$\min_\rho \widehat{\mathrm{DSMW}}_L^2(\mu_\rho, \mu).$$

The updated diagonal entries of $Q^k$ are here projected back to $\mathbb{R}_{\geq 0}$. As target GMM $\mu$, we use two-dimensional Gaussian mixtures with 100 components in the form of $28 \times 28$ MNIST digits (LeCun et al., 1998), which are calculated by employing the EM algorithm (Dempster et al., 1977) to the pixel intensities, see Figure 4. Applying Adam with step size 0.03 for 200 iterations and 20 random initializations, and choosing $L = 100$ and $\sigma = 1$ (pixel), we quantize the inputs by 50-component GMMs. Each gradient descent cycle requires less than a second GPU time. The resulting densities on $[0, 28]^2$ are shown in Figure 4.

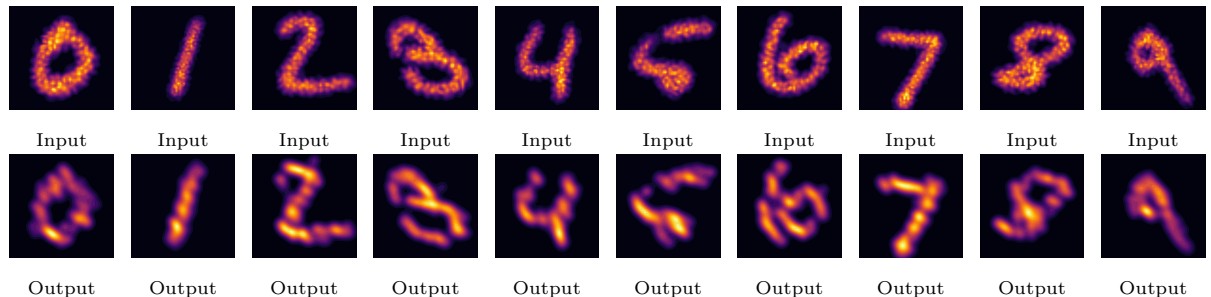

Figure 4: Quantization of input GMMs with 100 components (top) to GMMs with 50 components (bottom).

## 5.4 Barycenter of Gaussian mixtures

Barycenters are generalized Fréchet means, which in the context of optimal transport are defined as measure-valued midpoints between multiple inputs. For general measures, Wasserstein barycenters (Agueh & Carlier, 2011) and sliced Wasserstein barycenters (Bonneel et al., 2015) as well as, for GMMs, mixture Wasserstein barycenters (Delon & Desolneux, 2020) are studied. The latter have applications in computational biology (Lin et al., 2023) and domain adaptation (Montesuma et al., 2024). In more detail, for $\mu_1, \ldots, \mu_I \in \mathrm{GMM}_d(K)$ and $\lambda \in \Delta_I$, a MW barycenter is the solution to

$$\inf_{\mu^* \in \mathrm{GMM}_d(\infty)} \sum_{i=1}^{I} \lambda_i \, \mathrm{MW}^2(\mu^*, \mu_i). \tag{16}$$

The infimum of (16) is always attained, where the components $\mu^{*,k_1,\ldots,k_I} \in \mathrm{GMM}_d(1)$ of the minimizer $\mu^*$ are themselves Wasserstein barycenters between $\mu_1^{k_1}, \ldots, \mu_I^{k_I}$, i.e.,

$$\mu^{*,k_1,\ldots,k_I} := \operatorname*{arg\,min}_{\mu^\dagger \in \mathcal{P}_2(\mathbb{R}^d)} \sum_{i=1}^{I} \lambda_i \, \mathrm{W}_2^2(\mu^\dagger, \mu_i^{k_i}), \tag{17}$$

which can be calculated analytically (Delon & Desolneux, 2020). The weights of these closed-form Wasserstein barycenters can be computed by solving a linear program, where at most $(I-1)K - 1$ weights are non-zero.

In the style of the MW barycenter problem (16), and as proof-of-concept, we initially consider (fixed-component) DSMW barycenters $\mu^* \in \mathrm{GMM}_d(K^I)$ with components $\mu^{*,k_1,\ldots,k_I}$ from (17) solving

$$\inf_{\mu^* \in \mathrm{GMM}_d(K^I)} \sum_{i=1}^{I} \lambda_i \, \mathrm{DSMW}^2(\mu^*, \mu_i). \tag{18}$$

Referring to Section 5.3, we solve (18) using stochastic gradient descent with automatic differentiation, where we parametrize $\mu^*$, whose components are fixed, by

$$\mu_w := \sum_{k_1=1}^{K} \cdots \sum_{k_I=1}^{K} \frac{\exp(w^{k_1,\ldots,k_I})}{\sum_{\ell_1=1}^{K} \cdots \sum_{\ell_I=1}^{K} \exp(w^{\ell_1,\ldots,\ell_I})} \, \mu^{*,k_1,\ldots,k_I}, \qquad w \in \mathbb{R}^{K \times I}.$$

For comparison, we adapt the MW interpolation task[2] first presented in (Delon & Desolneux, 2020). Approximating (18) using $\widehat{\mathrm{DSMW}}_L^2$ with 100 random directions, stochastic gradient descent finds the optimal weights parametrized by $w$ after 10 iterations with a step size of 0.03. The results presented in Figure 5 show that MW and DSMW both enable a meaningful smooth interpolations despite their distinct geometry. A direct comparison convey the impression that the DSMW barycenter is less related to the prominent features of the inputs. Especially, in the middle, we nearly obtain unimodal GMMs. Note that the aim of

---

[2]https://github.com/judelo/gmmot

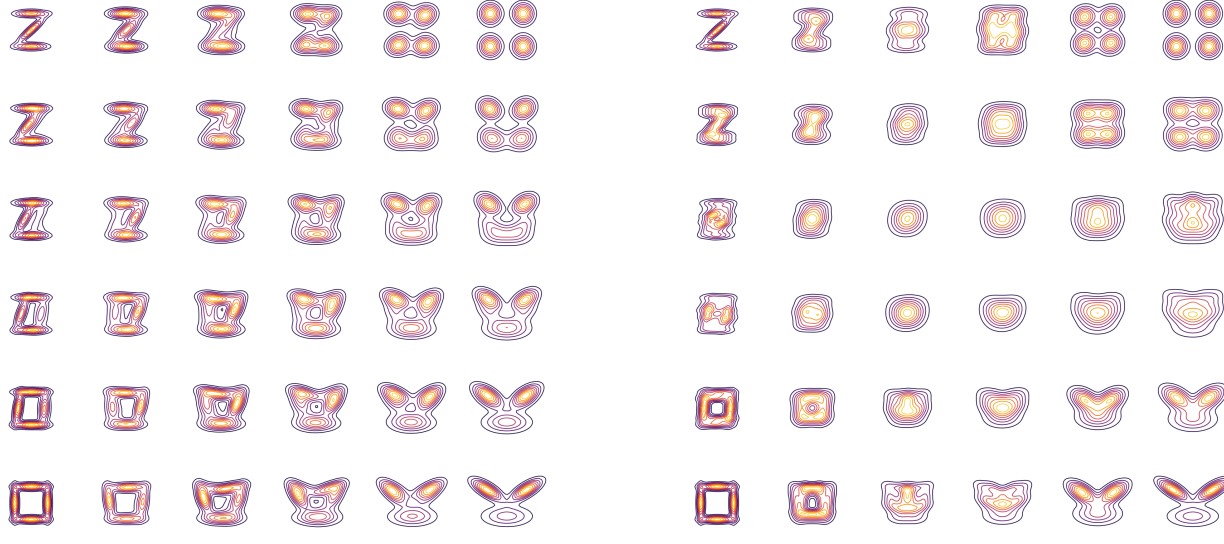

MW barycenter                                               Fixed-component DSMW barycenter

Figure 5: Bilinear interpolation between GMMs at the corners via MW barycenters and fixed-component DSMW barycenters based on the example from (Delon & Desolneux, 2020). Both methods show reasonable, smooth interpolations between the inputs.

this toy example is only to demonstrate that the DSMW distance allows efficient barycenter computations using stochastic gradient descent. The exploration of the quality, especially compared to other optimal transport-based barycenters, is left for future research.

Despite the underlying sparse structure of the MW barycenters, computation becomes increasingly costly beyond toy examples due to the necessary minimization over the weights of $K^I$ Gaussian components (Lin et al., 2023). Based on the parametrization $\mu_\rho \in \mathrm{GMM}_d(K^*)$ in (15) with $\sigma = 0.3$ (pixel) and an arbitrary component number $K^* \in \mathbb{N}$, we may alternatively use the Adam scheme with automatic differentiation to approximate a (free-component) DSMW barycenter by minimizing

$$\min_\rho \sum_{i=1}^I \lambda_i \, \widehat{\mathrm{DSMW}}_L^2(\mu_\rho, \mu_i). \tag{19}$$

In the second experiment, we aim to compute the barycenters (19) between five MNIST samples (LeCun et al., 1998) per digit. The MNIST samples themselves are approximated by 10-component GMMs using the EM method. We refer to Figure 6 for the density contour plots of the inputs. For the barycenter computation, we choose $K^* = 100$, $L = 100$, and $\lambda = \frac{1}{5} \mathbf{1}_5$. Taking 200 Adam gradient descent steps with step size 0.03, where the descent is started from 10 random initial GMMs, we obtain the free-component DSMW barycenters in Figure 6. Along the results, we further display $\mathrm{SW}_2$ barycenters estimated by minimizing

$$\min_{x_n \in \mathbb{R}^2} \sum_{i=1}^I \lambda_i \, \widehat{\mathrm{SW}}_{2,L}^2 \Big( \frac{1}{N} \sum_{n=1}^N \delta_{x_n}, \tilde{\mu}_i \Big),$$

where $\tilde{\mu}_n$ are empirical measures obtained from 10000 samples from $\mu_n$. Similarly to the DSMW barycenter, we choose $N = 100$, $L = 100$, $\lambda = \frac{1}{5} \mathbf{1}_5$ and apply 200 Adam gradient descent steps with step size 0.03 and 9 restarts. The results are also visualized in Figure 6, where similar shapes emerge for both ansätze. The $\mathrm{SW}_2$ barycenters seem to be more noisy. For the described setting, 200 gradient descent steps require around 1s GPU time for DSMW and around 5s for $\mathrm{SW}_2$. Besides the computational speed-up, the main benefit of using our DSMW distance is that the DSMW barycenter is an actual undegenerated GMM, where the $\mathrm{SW}_2$ barycenter only yields a (degenerated) point cloud. Note that the shape of the DSMW and $\mathrm{SW}_2$ barycenter

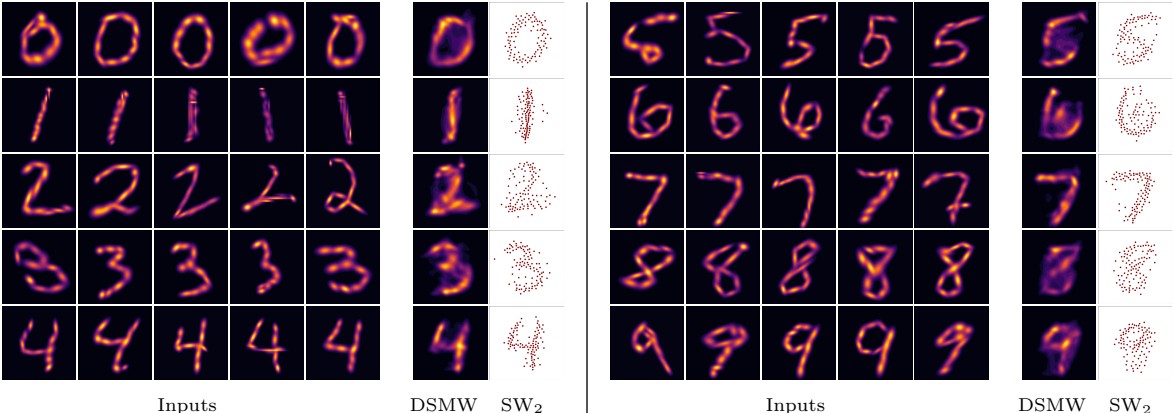

Figure 6: DSMW and $SW_2$ barycenters. Given an input set of five two-dimensional Gaussian mixtures in the form of MNIST digits (left: 0-4, right: 5-9), we display DSMW (in the form of a GMM density) and $SW_2$ barycenters (in the form of points). The DSMW barycenters are actual undegenerated GMMs, where the $SW_2$ barycenters are only (degenerated) point clouds.

are closely related, and that the DSMW barycenter can thus be interpreted as smoothed version of the $SW_2$ barycenter.

## 6 Conclusion

In this work, we introduce two novel sliced versions of the MW distance for GMMs, namely, the MSW and the DSMW distance. Both variants significantly reduce the computational burden while relying on the Euclidean geometry of the underlying domains. Our theoretical results established connections between MSW, DSMW, and MW as well as relations to $SW_2$. The latter especially ensures meaningful convergence and topological properties. Through extensive numerical experiments, we demonstrated the efficiency of our approach in various applications, namely, perceptual data comparison, unsupervised clustering, GMM quantization, and GMM barycenter computation.

Additionally, our framework can be readily adapted to so-called max-sliced distances, where the integral over the sphere is replaced by taking the maximum over the sphere. The original max-sliced Wasserstein distance has been shown to provide a meaningful alternative to the sliced Wasserstein distance by focusing on the most discriminative direction (Deshpande et al., 2019). This might be beneficial for gradient-based optimization. Moreover, the max-sliced distance is 1-strongly equivalent to the Wasserstein distance on $GMM_d(1)$ (Bayraktar & Guo, 2021). Consequently, replacing the sliced distance in the MSW construction with the max-sliced variant, we can strengthen the shown weak equivalence to an actual strong equivalence. Furthermore, it might be interesting to investigate metric equivalences with respect to the slicing techniques in (Nguyen & Mueller, 2025; Nguyen et al., 2025).

Beyond GMMs, the MW distance has been extended to mixtures of non-Gaussian probability measures (Alvarez-Melis & Fusi, 2020; Bing et al., 2022; Dusson et al., 2023). While our focus has been on Gaussian mixtures, an interesting direction for future research is the extension of our framework to other relevant non-Gaussian mixture models, such as Dirichlet mixtures (Pal & Heumann, 2022; Martin et al., 2024). Since many real-world datasets exhibit non-Gaussian characteristics (Luzi et al., 2023), developing Wasserstein-type distances tailored to these distributions would further broaden the applicability of our approach. Beyond the MW distance, a sliced extension to isometry-invariant optimal transport between Gaussian mixtures (Salmona et al., 2024; Beier et al., 2025) might be of interest.

### Author Contributions

Both authors—MP and RB—have equally contributed to this paper.

**Acknowledgments**

MP gratefully acknowledges the financial support by the German Research Foundation (DFG), GRK2260 BIOQIC project 289347353.

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
