# OpenReview forum: "Slicing the Gaussian Mixture Wasserstein Distance"
_TMLR — Accepted by TMLR_

### Review · Reviewer_cDSz · 2025-05-26

**Summary Of Contributions:**

The work proposes two new distances on Gaussian mixtures; the mixture sliced Wasserstein distance (MSW) and the double-sliced mixture Wasserstein distance (DSMW). This addresses the problem of expensive computation of the mixture Wasserstein (MW) distance, previously proposed to compute distances between Gaussian mixtures, which arises due to (i) poor scalability in space dimension $d$ and (ii) number of mixture components $K$. The MSW addresses the former problem, and the DSMW addresses both problems, at the cost of being a weaker metric. Theoretical results show the weak equivalence between MW, MSW, SMW and DSMW, strong equivalence of SMW and DSMW and the weak topology induced by SMW and DSMW. Experiments demonstrate the computational benefits of using the newly proposed distances with various applications.

**Audience:**

Yes

**Broader Impact Concerns:**

The work is mostly theoretical, and there are no immediate ethical concerns.

**Claims And Evidence:**

Yes

**Requested Changes:**

- In the proofs of Theorem 3.1 and 3.4, I believe there is a misprint in the notation of the optimal transport plan, where sometimes the notation $\pi^{n, \theta}\_{k\_0, k\_1}$ is used instead of $\gamma^{n, \theta}\_{k\_0, k\_1}$.
- There is a misprint in the proof of Theorem 4.2: in the second-to-last line of the equation, $1/L$ should be inside the parentheses, not outside.
- Misprint in page 13: "Cholseky decomposition" -> "Cholesky decomposition".
- I am not sure what the phrase "fitted GMM with $k$ components and a one with $k+1$" in the caption of Figure 2 means.
- Please add a label for the y-axis in Figure 2, as it can be confusing what is being displayed at first glance. My understanding is that these are the {MW, MSW, DSMW} distances between the true Gaussian mixture and the predicted one using EM. I would recommend adding a y-axis label "GMM Distance" or something to that effect, and explain in the caption or main body how the distances are computed (e.g. the blue curve displays MW distance between the true model and the predicted model, etc...).
- Similarly, please add labels for the x and y-axes in Figure 3.
- In the conclusion, please be clearer about the sentence "while preserving the geometric properties". What geometric properties are the authors referring to?

**Strengths And Weaknesses:**

__Strengths:__
- Overall, the paper is easy to follow; the objectives are clear and the main results are presented nicely, aided by a summarising figure in Figure 1.
- Claims are backed by rigorous proofs, which I have checked and seem correct. These theoretical results are also insightful about the newly proposed metrics.
- Numerical experiments are extensive and show the benefits and properties of the proposed distances both computationally and in terms of their various applications.

__Weaknesses:__
- Clarity on some of the experiments can be improved (see more below).

---

> ### Author Response · Authors · 2025-06-11
> **Revision of Paper4653**
>
> Thanks a lot for your review and your constructive feedback!
> We have revised our manuscript accordingly,
> where all changes are marked in blue.
> Please find our detailed answers to your questions below.
>
> - We fixed the typo regarding the transport plan.
> - We corrected the equation.
> - We fixed the typo in Cholesky.
> - We adapted the caption of Figure 2
>   to make clearer
>   that we compute the GMM distances
>   between the fitted models $\mu_k$ and $\mu_{k+1}$.
>   If the distance between these becomes zero,
>   the new component only yield a reparametrization of the former model.
>   In the experiment,
>   the first zero distance corresponds exactly
>   to the correct cluster number.
> - We added a y-axis label to Figure 2
>   and adapted the caption and running text as mentioned above.
>   Note that we do not compare the fitted GMMs with the ground truth GMM.
>   We hope that the extended text is more understandable.
> - We added axes labels to Figure 3.
> - We adapted the text to clarify that
>   we exploit the Euclidean geometry of the underlying domains
>   for the definition of MSW and DSMW.

---

### Review · Reviewer_Mo6r · 2025-06-02

**Summary Of Contributions:**

This paper addresses the computational limitations of the Mixture Wasserstein (MW) distance between Gaussian Mixture Models (GMMs) by introducing slicing-based approximations.

The proposed alternatives—Mixture Sliced Wasserstein (MSW), Sliced Mixture Wasserstein (SMW), and Double-Sliced Mixture Wasserstein (DSMW)—are theoretically analyzed and shown to be metrics.

Theoretical results regarding their weak and strong equivalences with MW and Sliced Wasserstein (SW) distances are provided.

Empirical applications include clustering, perceptual similarity assessment, quantization, and barycenter computation.

**Audience:**

Yes

**Broader Impact Concerns:**

Not applicable to this paper in my opinion.

**Claims And Evidence:**

Yes

**Requested Changes:**

**[Critical to securing my recommendation for acceptance & Comments to strengthen the work in my view]**

### [C1]
Propositions except 3.6 utilized the `compact parameter sets', but the explanation on the importance of this property is insufficient. Is this assumption just for proofing theorems? In contrast, do ML tasks have some merits on this?

### [C2]

Section 4.1 and 4.2 seems to be the main contribution and Section 4.3 - 4.6 show application examples; thus some main message is now degraded.

Some comment for this:
- A possible way is dividing the section.
- Do we real data examples on high-dimensional domains?
- For clarity, give an explanation to stop MC computation by comparing L and L + 10 results (i.e., 10 is a kind of magic number for eaders).
- In Eq. (15), $\mathrm{W}$ means $W_p$?
- Please justify using $k=4$ examples in Figure 2: the results are insightful, but readers may not follow the importance and reason to show this result here.
- Please clarify the advantages of DSMW in Figure 5 and Figure 6; some readers may be difficult to follow this qualitative comparisons using figures (basically, this just shows some gaussian crowds and points, digits).

### [C3]
Please carefully write equations to make the discussions clearer.

The below are examples:
- $\mathrm{W}_p$ and $W_p$ are mixed.
- [above Eq. (3)] What is the meaning of $\subset\subset$?
- [below Eq. (3)] $SW(\mu_n,\mu)$ and $W(m_n, mu)$ mean $SW_p$ and $W_p$ for any $p$? (This paper did not define SW or W without a subscript.)
- [Eq. (7)] What is the exact definition of this sigma ($\sum_{k_0, k_1=1}^{K_0, K_1}$).
- [Eq. (8)] Where from is this equation? Delon & Desolneux 2020?
- [Eq. (9)] What is the exact definition of $\leq$ between metrics?
- [Proof of Theorem 4.2] Should 1/L in line 3 be in the parentheses? (to extract 1/sqrt(L))

**Strengths And Weaknesses:**

**Strengths**
- [S1] Clear Computational Advantage: MSW and DSMW drastically reduce runtime compared to MW, as demonstrated in Table 1.
- [S2] Solid Theoretical Foundation: Theoretical results (Figure 1, Theorems 3.1–3.4) provide important insight into metric properties and equivalence relationships.
- [S3] Paper is well-organized and easy to follow structurally.

**Weaknesses**
- [W1] Limited explanation of compact parameter assumptions: Many key theoretical results rely on the assumption of compactness, but its implications and relevance to real ML applications are not sufficiently discussed, related to [C1].
- [W2] Applications are somewhat illustrative rather than practical: Section 4 showcases several applications, but they feel scattered, limited in diversity, and do not fully leverage high-dimensional real-world datasets, related to [C2].
- [W3] Notational inconsistencies and unclear formulations: Some equations (e.g., (3), (7), (8)) lack clarity or proper references; inconsistent use of notations (e.g., $W_p$ and $\mathrm{W}_p$) also hinders readability, related to [C3].

---

> ### Author Response · Authors · 2025-06-11
> **Revision of Paper4653**
>
> Thanks a lot for your review and your constructive feedback!
> We have revised our manuscript accordingly,
> where all changes are marked in blue.
> Please find our detailed answers to your questions below.
>
> **(C1) Compactness Assumption**
> In the current version,
> the compactness of the parameter sets is just needed to proof the theorems.
> Induced by your comment,
> we have perused our proofs,
> especially of Theorem 3.4,
> to overcome this assumption.
> Although we conjecture that the statement remain true,
> the current proof technique does not allow to skip this.
> From the view of ML tasks,
> the compactness assumption is unproblematic
> as long as
> we consider GMMs
> whose variances and means are bounded.
> In this case,
> the parameter set can be chosen as closed ball with sufficient radius.
> We added this discussion to the updated manuscript.
>
> **(C2) Numerical Implementation and Experiments**
> - As suggested,
>   we have divided the numerical implementations
>   and the numerical experiments
>   into two separate sections.
> - We agree that our application are mostly illustrative.
>   Our aim has been to collect applications of the MW distance
>   that might benefit from our new distances.
>   In the moment,
>   the actual application of DSMW
>   for real-world applications is left for future studies.
>   The current version,
>   however,
>   contains one possible application
>   involving high-dimensional domains.
>   More precisely,
>   the perceptual metrics in Section 5.2 are based on GMMs
>   in a 2048-dimensional feature space
>   for real images.
> - We choose the index shift 10 to stop the Monte Carlo iteration
>   only if the estimate does not change over a few iterations.
>   This explanation has been added after Equation (13) and (14).
> - Yes, we fixed the typo in Eq. (15).
> - We agree
>   that $k=4$ is somehow arbitrary.
>   Therefore,
>   we increased the number of components in the last two experiments.
>   We also improved the representation of the results.
> - We now clearly state
>   that Figure 5 should merely show
>   that the computation of the DSMW barycenter using stochastic gradient decent is possible
>   and does not directly show any further advantages of DSMW.
>   For the experiment behind Figure 6,
>   the main benefit is that
>   DSMW yields an undegenerated GMM barycenter,
>   whereas SW only yields a (degenerated) point cloud.
>   This advantage is now more clarify stated in the caption and running text.
>
> **(C3) Further Comments**
> Thanks for your very helpful suggestions.
> We simplified the notations accordingly
> (remove the symbol $\subset\subset$ for compact subsets
> and write out the double sums),
> remove inconsistencies
> (add indices and arguments to W and SW),
> fixed the typos,
> and adapted the citations.

---

### Review · Reviewer_XP7B · 2025-06-03

**Summary Of Contributions:**

The paper proposes a family of sliced approximations to the Mixture Wasserstein (MW) distance for Gaussian‑mixture models (GMMs).
By repeatedly projecting the mixtures onto one‑dimensional subspaces, the authors get two computable metrics, i.e., Mixture Sliced Wasserstein (MSW) and Double‑Sliced Mixture Wasserstein (DSMW).
These metrics (i) retain the distance properties of MW, (ii) come with Monte‑Carlo estimators whose cost scales linearly in dimension and number of components, and (iii) dramatically reduced the computational cost of OT‑based tasks such as clustering, model selection, and barycenter computation.

**Audience:**

Yes

**Claims And Evidence:**

Yes

**Requested Changes:**

1. Discuss the the trade-offs between sliced Wasserstein distance and entropically regularized Wasserstein distances, e.g. comparing MSW with mixtures of sinkhorn distances. If feasible, also extend the experimental section to include this comparison.
2. To show the scalability of the proposed method, extend the synthetic benchmarks to higher dimension d and component K in the experiments.
3. To show how the number of projections $L$ needed scales with dimension d and component K, in the runtime analysis, include the time used for each projection, and the number of projections $L$ used to reach the convergence criteria, together with the total time used for each method.

**Strengths And Weaknesses:**

Strengths:
1. The slicing approach provides a simple yet theoretically grounded means of making MW computations feasible for high-dimensional GMM applications.
2. The paper rigorously analyzes the relationships between WM, MSW, SMW, and DSMW, including their inequality structure, equivalence properties, and topological behaviors.
3. Runtime, clustering, perceptual quality, and barycenter examples illustrate both efficiency and utility.
4. The theoretical derivations and algorithmic descriptions are clear and well-organized.

Weaknesses:
1. The paper offers limited discussion of alternative approaches to reduce the computational cost of the Wasserstein distance. For instance, a comparative analysis with entropically regularized OT methods such as Sinkhorn distances would be informative.
2. While Theorem 4.1 and 4.2 showed the expected absolute error of SW and DSWM, since the right hand side of the inequalities, which are interpreted as standard deviation terms, increase as dimension d and component K increase, the scalability of $L$ required to reach convergence of the algorithm is still not clear.
3. In the runtime analysis, DSMW demonstrates a significant empirical speedup over MW; however, the scalability of the proposed methods remains unclear due to the limited range of parameter settings explored in the experiments.

---

> ### Author Response · Authors · 2025-06-11
> **Revision of Paper4653**
>
> Thanks a lot for your review
> and your constructive feedback!
> We have revised our manuscript accordingly,
> where all changes are marked in blue.
> Please find our detailed answers to your questions below.
>
> **Entropic Regularization**
> We extended the runtime comparison (Section 4.2) accordingly.
> More precisely,
> we speed-up the computation of the outer OT problem behind MW
> by using an entropic regularization
> and the Sinkhorn algorithm
> from the POT library.
> The inner OT problems are still solved analytically.
> The run-times for the additional entropic version of MW
> are reported in Table 1.
> This adaption only yields a minor acceleration
> showing that the inner OT problems are the major bottleneck.
> We observed a similar minor speed-up
> if the outer OT behind MSW is regularized.
> Due to the analogy to MW,
> we however did not add an in-depth example for MSW.
>
> **Scalability**
> We extended our runtime analysis in Section 4.2
> by adding Table 2 and Table 3,
> where the performance for high component numbers
> and high-dimensional domains is studied.
> In both experiments,
> we observe
> that MSW is stretched to one's limits
> whereas DSMW is still computed in seconds.
> We did not extend our analysis beyond the new experiments
> since we ran into memory issues
> due to size or the number of required covariance matrices.
>
> **Runtime Analysis**
> We added the average number of projections
> until convergence
> to the new experiments
> in Table 2 and Table 3.
> Overall,
> it seems
> that DSMW requires much less projections than MSW.
> Especially,
> in for high component numbers,
> we observe convergence already after using 15 projections.
> We conjecture
> that
> the reason for this behavior is based on
> the random procedure for the GMM generation;
> for very high component numbers,
> the generated GMMs look nearly alike.
> Since MSW compares the single components,
> this effect do not lead to a computational speed-up.

---

### Decision · Action_Editor_gxvS · 2025-08-06

**Recommendation:** Accept as is

**Additional Comments:**

This paper proposes slicing-based approximations, MSW, SMW, and DSMW, for the Mixture Wasserstein distance between Gaussian Mixture Models, addressing key computational bottlenecks. Theoretical analysis and new experiments (e.g., entropic MW baseline, high-dimensional tests) support the scalability and practical value of the approach. The presentation is clear, and the paper is technically sound.

While large-scale validation and approximation of inner OT steps remain open, the contribution is reasonable, and it can be useful for density modeling, domain adaptation, and dataset auditing. All reviewers agree on publication, and I also recommend acceptance.

**Audience:**

Yes

**Audience Explanation:**

The sliced Wasserstein distance is a key topic in optimal transport research and attracts strong interest from the OT community.

**Claims And Evidence:**

Yes

**Claims Explanation:**

The proposed claims are well justified by both theory and experience.